# Tensor Train Diffusion:
# Leveraging Low-Rank Structures for High-Dimensional Score-Based Sampling

**Robert Gruhlke** [* 1]   **Julius Berner** [2]   **David Sommer** [3]   **Lorenz Richter** [* 4 5]

## Abstract

Diffusion models offer a powerful framework for sampling from complex probability densities by learning to reverse a noising process. A common approach involves solving for the time-reversed stochastic differential equation (SDE), which requires the score function of the evolving sample distribution. The logarithm of this distribution's density is governed by a Hamilton-Jacobi-Bellman (HJB) type partial differential equation (PDE). However, current methods for solving this PDE, such as PINNs or trajectory-based techniques, often suffer from long training times and significant sensitivity to hyperparameter tuning. In this work, we introduce a novel and efficient solver for the underlying HJB equation based on the functional tensor train (FTT) format. The FTT representation leverages latent low-rank structures to efficiently approximate high-dimensional functions, enabling both model compression and rapid computation. By integrating this efficient representation with a backward-in-time iterative scheme derived from backward stochastic differential equations (BSDEs), we develop a fast, robust and accurate sampling method. Our approach overcomes primary bottlenecks of existing techniques, enabling high-fidelity sampling from challenging target distributions with improved efficiency.

## 1 Introduction

Sampling from a complex, high-dimensional probability density

$$p_{\text{target}} = \frac{\rho_{\text{target}}}{\mathcal{Z}}, \tag{1}$$

---

[*]Equal contribution  [1]Freie Universität Berlin  [2]NVIDIA  [3]WIAS  [4]dida  [5]Zuse Institute Berlin. Correspondence to: Robert Gruhlke <r.gruhlke@fu-berlin.de>, Lorenz Richter <richter@zib.de>.

*Proceedings of the $43^{rd}$ International Conference on Machine Learning*, Seoul, South Korea. PMLR 306, 2026. Copyright 2026 by the author(s).

where the unnormalized density $\rho_{\text{target}}$ can be evaluated pointwise but the normalizing constant $\mathcal{Z}$ is intractable and no samples from $p_{\text{target}}$ are available, constitutes a central challenge in modern machine learning, statistics, and the physical sciences. An especially versatile and powerful class of methods for this task is based on dynamical measure transport, which generates samples from the target distribution by evolving initial samples from a simple reference distribution – such as a standard Gaussian – along trajectories defined by stochastic or ordinary differential equations. One of the most successful paradigms within this framework is the time reversal of a noising SDE, forming the foundation of *diffusion models* (Ho et al., 2020; Song et al., 2021). In this approach, a forward SDE – typically chosen as an *Ornstein-Uhlenbeck process* – progressively transforms the target distribution into a simple prior by injecting noise over time. The central insight is that reversing this noising process yields a generative SDE that transports samples from the prior back to the target distribution. A classical result (Nelson, 1967; Anderson, 1982) establishes that the drift of this time-reversed SDE is determined by the *score function*, $\nabla \log p(\boldsymbol{x}, t)$, where $p(\cdot, t)$ denotes the density of the forward process at time $t$. Thus, the sampling problem reduces to accurately approximating this time-dependent score function. While diffusion models were originally developed for the generative-modeling setting, in which the score is learned from samples via denoising score matching, here we adopt only the time-reversal mechanism: the score must instead be obtained without samples, directly from the unnormalized density $\rho_{\text{target}}$.

Since the densities of SDEs are governed by the *Fokker-Planck equation*, one can employ the *Hopf-Cole* transform to relate the evolution of the log-density to a *Hamilton-Jacobi-Bellman* (HJB) equation (Berner et al., 2024). However, solving this PDE is notoriously challenging, particularly in the high-dimensional settings typical in practical applications, which are intractable for traditional mesh-based approaches. In the context of diffusion models, there exist attempts to approximate solutions to such HJB equations with neural networks, for example via *physics-informed neural networks* (PINNs) (Sun et al., 2024; Shi et al., 2024b). However, the optimization of neural networks with stochastic gradient descent (SGD) requires a large number of (potentially

costly) evaluations of $\rho_{\text{target}}$ and relies on automatic differentiation to compute the higher-order derivatives appearing in the PDE. This leads to long training times, sensitivity to hyperparameters, and convergence to local minima (Krishnapriyan et al., 2021; Shi et al., 2024a; Xu et al., 2025).

In this work, we depart from neural network-based PDE solvers and propose a novel approach based on *functional tensor train* (FTT) representations. Tensor trains are particularly well-suited for this setting, as they can efficiently represent high-dimensional functions by exploiting latent low-rank structures, enabling both substantial model compression and fast computation. Moreover, the tensor train format is naturally compatible with backward-in-time regression schemes, which can be derived from the connection between HJB equations and backward stochastic differential equations (BSDEs). This yields an efficient implementation that circumvents the instabilities and complexities associated with SGD-based optimization. Our main contributions can be summarized as follows:

- We introduce *Tensor Train Diffusion* (TTD)[1], a novel solver for HJB equations that leverages connections between diffusion-based sampling, HJB equations, and BSDEs to achieve a favorable trade-off between computational cost and accuracy.

- We demonstrate the flexibility of TTD by incorporating different basis functions – including Legendre polynomials, B-splines, and Fourier bases – and substantially improve robustness compared to previous tensor-train-based PDE solvers, effectively handling sampling trajectories outside the primary training domain.

- We show that TTD can be used to sample from complex, multimodal target distributions of varying dimensionality, outperforming existing diffusion-based samplers in both speed and accuracy.

## 1.1 Related work

**Sampling from unnormalized densities.** Drawing samples from a density, specified up to the normalizing constant, is a crucial task in a wide range of computational sciences, ranging from molecular dynamics, to lattice field theory, to Bayesian statistics (Gelman et al., 2013; Zhang et al., 2023). Due to the great interest, a large amount of literature focused on developing corresponding sampling methods, often based on variants of *Markov chain Monte Carlo* (MCMC) and *Langevin dynamics* (Neal, 2001; Chopin, 2002; Del Moral et al., 2006). However, such algorithms typically require substantial tuning and long runtimes to sample from high-dimensional distributions with well-separated modes (Latuszyński et al., 2025; Brooks et al., 2011).

[1]The code is available at https://github.com/robertgruhlke/TTD.

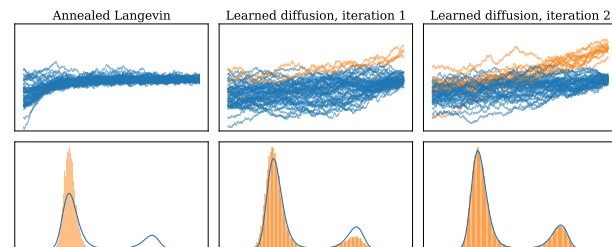

*Figure 1.* Overview of the proposed TTD method. From left to right: Annealed Langevin dynamics serve as an initialization for our approach. By learning the score function along relevant trajectories, TTD iteratively refines the sampling process. As a result, new modes are discovered and the quality of samples improves. The histograms illustrate the terminal samples of the trajectories in comparison to the target density.

**Diffusion-based samplers.** To combat these issues, variational inference methods propose to turn the sampling problem into an optimization problem over a tractable family of distributions. Translating the success of diffusion models to sampling problems, recent works proposed to use parametrized SDEs for these families (Richter & Berner, 2024; Berner et al., 2024; 2026; Vargas et al., 2024; Blessing et al., 2025; 2026a;b). Since the drifts of the SDEs are typically parametrized by neural networks that incorporate the gradient of the target density, such approaches can be viewed as a learnable corrections to Langevin dynamics, aiming to converge after finite trajectory lengths. However, this also leads to a significant amount of (potentially costly) density evaluations during SDE simulations, which are typically also needed for training. While such methods can outperform classical methods in terms of inference performance and efficiency, they suffer from a high upfront cost for optimizing the parameters of the SDE.

**Neural PDE solver.** Since the dynamics of SDEs are governed by Fokker-Planck equations, we can equivalently solve the corresponding PDE in order to optimize the variational family of SDEs. However, such high-dimensional PDEs are out of scope for traditional mesh-based methods, such as finite differences or finite elements, which suffer from the *curse of dimensionality*. There is a series of works proposing variants of physics-informed neural networks for this problem (Shi et al., 2024b; Albergo & Vanden-Eijnden, 2025; Sun et al., 2024). While they can scale to higher dimensions, they do not address the high training costs due to their reliance on evaluations of the target density and higher-order derivatives as well as high sensitivity to hyperparameters. Methods that are based on iterative diffusion optimization additionally suffer from large computational costs to repeatedly simulate diffusion trajectories (Nüsken & Richter, 2021; 2023).

**Tensor trains.** Many target densities of interest exhibit an inherent low-rank structure. To exploit this, we propose to approximate the drift using functional tensor trains (FTT, Os-

eledets (2013)) in their numerical realization, the *extended (functional) tensor train* format (Eigel et al., 2022; Strössner et al., 2024) is sometimes referred to as *spectral tensor trains* (Bigoni et al., 2016). The effectiveness of such representations has already been demonstrated in related contexts, for example in sampling problems connected to HJB equations (Gruhlke et al., 2026), as well as for the Fokker-Planck and HJB equations in Dolgov et al. (2012). Low-rank tensor formats enable efficient regression-based solvers with broad applicability; see Bachmayr (2023) for an overview. Moreover, they constitute a rigorous nonlinear approximation class, with detailed analyses provided in Kazeev & Schwab (2018); Ali & Nouy (2020a;b; 2021); Bachmayr et al. (2021); Griebel et al. (2023); Bachmayr (2023). In this work, we build on the ideas developed in Richter et al. (2021; 2024) and extend them to the sampling setting. Inspired by Bouchard & Touzi (2004); Gobet et al. (2005); Huré et al. (2020), we notice that the Fokker-Planck equation is the linearization of a Hamilton-Jacobi-Bellman equation under the Hopf-Cole transform, such that the optimal drift can be written in terms of backward SDEs (BSDEs) that lead to simple regression problems. Together with the TT parametrization, we show that this reformulation of the PDE leads to an efficient and scalable sampler.

**Notation.** We define time-inversion as $\overleftarrow{f}(t) := f(T - t)$. For other notation, including our tensor notation, we refer to Table 1 in the appendix.

## 2 Diffusion-based sampling

In this section, we demonstrate how learned stochastic processes can be employed to sample from the target distribution and introduce a stochastic algorithm that formalizes the associated learning task.

### 2.1 Time-reversal and the related Hamilton-Jacobi-Bellman PDE

Our sampling algorithm for $p_{\text{target}}$ builds on the idea of reversing a noising process that approximately evolves towards an easy-to-sample distribution $p_{\text{prior}}$ at terminal time $T$ (Song et al., 2021; Berner et al., 2024). To this end, we consider the process

$$\mathrm{d}Y_s = -\overleftarrow{f}(Y_s, s)\,\mathrm{d}s + \overleftarrow{\sigma}(s)\,\mathrm{d}W_s, \quad Y_0 \sim p_{\text{target}}, \quad (2)$$

Unlike classical diffusion models with access to samples from $p_{\text{target}}$, the noising process cannot be simulated. The process $Y$ therefore serves only as an auxiliary analytical construct and is not part of the algorithm. The corresponding sampling dynamics, which can be simulated, is given by

$$\mathrm{d}X_s^u = \left(f + \sigma u\right)(X_s^u, s)\mathrm{d}s + \sigma(s)\mathrm{d}W_s, \quad X_0^u \sim p_{Y_T}, \quad (3)$$

where the objective is to learn a function $u$ that reverses the dynamics of (2), ensuring that $X_T^{u^*} \sim p_{\text{target}}$. It is well known that the optimal solution is given by the (scaled) *score function* $u^* = \sigma^\top \nabla \log \overleftarrow{p}_Y$ (Nelson, 1967; Anderson, 1982). Moreover, as shown in Berner et al. (2024), this function can be equivalently characterized via the following Hamilton-Jacobi-Bellman (HJB) PDE.

**Lemma 2.1** (HJB PDE for the log-density). *Let* $V := -\log \overleftarrow{p}_Y$. *Then,* $V(\cdot, T) = -\log p_{\text{target}}$ *and*

$$\partial_t V = -\tfrac{1}{2}\,\mathrm{Tr}(\sigma\sigma^\top\nabla^2 V) - f \cdot \nabla V \\ + \mathrm{div}(f) + \tfrac{1}{2}\big\|\sigma^\top\nabla V\big\|^2. \quad (4)$$

It is important to note that the boundary term may be replaced by its unnormalized version $-\log \rho_{\text{target}}$. This substitution merely shifts the PDE solution by an additive constant, which is irrelevant for the optimal control since only the gradient of the solution enters.

While equation (4) formally provides a recipe for approximating the log-density and, consequently, the score function, it is well known that solving PDEs in high dimensions is notoriously challenging. In the following, we therefore introduce a procedure that combines Monte Carlo estimation with suitable function approximation to efficiently handle high-dimensional settings.

### 2.2 BSDE representations and backward iterations

One approach to approximate certain nonlinear high-dimensional PDEs relies on backward stochastic differential equations (Pardoux, 1998). Their main idea relies on Itô's formula, which, for a stochastic process $X^u$ as defined in (3), states that

$$V(X_T^u, T) - V(X_0^u, 0) = \int_0^T \sigma^\top \nabla V(X_s^u, s) \cdot \mathrm{d}W_s + $$

$$\int_0^T \big(\partial_t V + \tfrac{1}{2}\,\mathrm{Tr}(\sigma\sigma^\top\nabla^2 V) + (f + \sigma u) \cdot \nabla V\big)(X_s^u, s)\mathrm{d}s. \quad (5)$$

Now, assuming that $V$ solves the HJB PDE (4), we may equivalently write $\mathrm{BSDE}(V) = 0$ with

$$\mathrm{BSDE}(V) := V(X_0^u, 0) - V(X_T^u, T) \\ + \int_0^T h(u, \nabla V)(X_s^u, s)\,\mathrm{d}s + \int_0^T \sigma^\top\nabla V(X_s^u, s) \cdot \mathrm{d}W_s,$$

where for convenience we defined the nonlinear term

$$h(u, \nabla V) := \mathrm{div}(f) + \tfrac{1}{2}\big\|\sigma^\top\nabla V\big\|^2 + u \cdot \sigma^\top\nabla V. \quad (6)$$

Conversely, when replacing $V$ by an approximation $\widetilde{V}$, $\mathrm{BSDE}(\widetilde{V}) = 0$ can only hold if $V = \widetilde{V}$ almost surely, by uniqueness of the PDE. This motivates to consider loss functionals of the form

$$\mathcal{L}(\widetilde{V}) := \mathbb{E}\Big[\big(\mathrm{BSDE}(\widetilde{V})\big)^2\Big], \quad (7)$$

where the expectation is over different realizations of the process $X^u$, cf. E et al. (2017). In principle, $V$ can now be learned by minimizing the loss (7) w.r.t. $V$. However, since solving the minimization problem in its entirety is difficult, it is natural to decompose it into a sequence of smaller subproblems. Following the dynamic programming principle from optimal control theory (Fleming & Rishel, 2012), we thus partition the time horizon into disjoint intervals according to the grid $0 = t_0 < t_1 < \cdots < t_N = T$, and aim to compute $V$ on each interval separately. Our envisioned algorithm proceeds backward in time, beginning with the final interval $[t_{N-1}, t_N]$ and employing the terminal condition $V(\cdot, T) = -\log p_{\text{target}}$. Next, for each preceding interval, the terminal condition is given by the solution obtained at the right endpoint of the subsequent interval. In practice, this means that $V(\cdot, t_n)$ is replaced by its previously computed approximation $\widetilde{V}(\cdot, t_n)$ for $n = 1, \ldots, N-1$. For further details we refer to Algorithm 1 and Richter et al. (2021; 2024).

In practice, we need to discretize both the process (3) and the (stochastic) integrals appearing in the loss (7). Using the same time grid as before, we approximate the solution only at the grid points, i.e., we seek functions $\widehat{V}_n$ such that $\widehat{V}_n \approx V(\cdot, t_n)$ for $n = 0, \ldots, N-1$. To this end, we employ the Euler-Maruyama scheme

$$\widehat{X}^u_{n+1} = \widehat{X}^u_n + (f + \sigma u)(\widehat{X}^u_n, t_n)\Delta t + \sigma(t_n)\xi_{n+1}\sqrt{\Delta t}, \ (8)$$

where $\Delta t = t_{n+1} - t_n$ is the time step and $\xi_{n+1} \sim \mathcal{N}(0, \text{Id})$ are independent standard normal random variables.

We then define for each $n = 0, \ldots, N-1$ the discrete loss

$$\widehat{\mathcal{L}}_n(\widehat{V}_n) = \mathbb{E}\Big[\Big(\widehat{V}_n(\widehat{X}^u_n) + \sigma(t_n)^\top \nabla \widehat{V}_n(\widehat{X}^u_n) \cdot \xi_{n+1}\sqrt{\Delta t}$$
$$+ h^u_{n+1}\Delta t - \widehat{V}_{n+1}(\widehat{X}^u_{n+1})\Big)^2\Big],$$
$$(9)$$

where $h^u_{n+1} := h(u, \widehat{V}_{n+1})(\widehat{X}^u_{n+1}, t_{n+1})$. The loss (9) is obtained by discretizing its continuous version (7) by choosing the right endpoint of the deterministic integral and the left endpoint of the stochastic integral. As a next step, the expectation in (9) is discretized by $K \in \mathbb{N}$ sample sequences $(\widehat{X}^{u,(k)}_n)_n$ for $k = 1, \ldots, K$, generated from (8). Defining $\Sigma^{(k)}_n = \sigma(t_n)\xi^{(k)}_{n+1}\sqrt{\Delta t} \in \mathbb{R}^d$ and $y^{(k)}_{n+1} = \widehat{V}_{n+1}(\widehat{X}^{u,(k)}_{n+1}) - h^{u,(k)}_{n+1}\Delta t$, the loss in (9) can be approximated using the empirical loss

$$\widehat{\mathcal{L}}^K_n(\widehat{V}_n) := \frac{1}{K}\sum_{k=1}^K \Big([\text{Id} + \Sigma^{(k)}_n \cdot \nabla]\widehat{V}_n(\widehat{X}^{u,(k)}_n) - y^{(k)}_{n+1}\Big)^2.$$
$$(10)$$

Note the affine dependence in $\widehat{V}_n$, making it well-suited for regression-based methods. Moreover, up to discretization error, the loss has the convenient property that it is almost

---

**Algorithm 1** Approx. of HJB PDE and optimal control

**Input:** Initial parametric choice for the functions $\widehat{V}^{(0)}_n$ and educated guess for $\widehat{u}^{(0)}_n$ for $n \in \{0, \ldots, N-1\}$. Number of outer iterations $I$.
**Output:** Approximation of $V(\cdot, t_n) \approx \widehat{V}^{(I)}_n$ and $u^*(\cdot, t_n) \approx \widehat{u}^{(I)}_n$ for $n \in \{0, \ldots, N-1\}$.
**for** $i = 1$ **to** $I$ **do**
  Simulate $K$ samples of the discretized SDE $\widehat{X}^{u^{(i-1)}}$ according to (8).
  Choose $\widehat{V}^{(i)}_N = -\log \rho_{\text{target}}$.
  **for** $n = N-1$ **to** $0$ **do**
    Discretize $\widehat{\mathcal{L}}_n(\widehat{V}^{(i)}_n)$ as in (9) using Monte Carlo.
    Minimize the resulting $\widehat{\mathcal{L}}^K_n(\widehat{V}_n)$ from (10) and set $\widehat{V}^{(i)}_n$ to be the minimizer.
    Set $\widehat{u}^{(i)}_n := -\sigma^\top(t_n)\nabla\widehat{V}^{(i)}_n$.
  **end for**
**end for**

---

surely zero at the solution, additionally implying vanishing variance at the optimum. For further analysis we refer to Richter et al. (2024).

*Remark* 2.2 (Gradient approximation). A key feature of the loss (9) is its explicit dependence on both the function $\widehat{V}_n$ and its gradient $\nabla\widehat{V}_n$. Because the loss penalizes errors in both terms, minimizing it provides a direct incentive to accurately approximate both the function and its derivative. This is particularly advantageous, as the optimal control relies directly on this gradient.

*Remark* 2.3 (Error propagation). When decomposing the problem into subproblems, one must take into account the possible propagation of approximation errors. These arise because, in practice, the approximation $\widehat{V}_{n+1}(\cdot) \approx V(\cdot, t_{n+1})$, which serves as a "terminal condition" for computing the individual losses, is generally not exact (see Section 8.3.3 in Gobet (2016)).

*Remark* 2.4 (Solving the PDE along a random grid). Unlike traditional numerical methods for PDEs, BSDE-based approaches can be interpreted as operating on dynamically adaptive random grids, provided by the grid points $(\widehat{X}^{u,(k)}_n)_{k,n}$. This perspective highlights their potential to achieve dimension-free Monte Carlo convergence rates.

*Remark* 2.5 (Fewer target evaluations). A strategy we adopt later is to first approximate the (unnormalized) log-target $\rho_{\text{target}}$ via a simple regression task. The resulting surrogate model can then be employed both for potential Langevin initialization runs and within the outer loop of Algorithm 1. This approach has the advantage that only a potentially small number of target evaluations are required initially; see also Remark 3.2.

Remark 2.4 highlights an additional challenge in sampling applications: we only need to approximate $V$, and hence the

control $u$, accurately along the trajectories of $X^u$. However, during training we learn $X^u$ along one set of sampled trajectories, while the actual evaluation later takes place along potentially different trajectories. To address this mismatch, we propose an iterative learning scheme with an outer loop that begins from an informed initial guess $u^{(0)}$ (typically corresponding to Langevin dynamics) and, at each step, updates the control using the solution $u^{(i)}$ obtained in the $i$-th iteration. This iterative refinement is expected to gradually align the training and evaluation distributions, so that the trajectories used for learning become sufficiently close to those relevant for sampling. In this way, the procedure concentrates on the regions of the state space that are most important for the optimal sampling process; see Algorithm 1 and Figure 1 for an illustration. We note that in our numerical experiments in Section 4 the algorithm typically converges in less than 3 iterations. Furthermore, we refer to Appendix A.4.4 for strategies to extend the learned control in a principled way beyond the data domain.

## 3 Tensor trains as approximating functions

This section introduces low-rank tensor formats for approximating the functions $\widehat{V}_n$. We focus on functional tensor trains (FTTs, Oseledets (2013)); for related work on low-rank function approximation, see Ali & Nouy (2020a;b; 2021); Bachmayr et al. (2021); Griebel et al. (2023); Bachmayr (2023). The FTT format is a nonlinear approximation class that exploits separability in the coupling of input variables. Its numerical realization via tensor trains yields a Riemannian manifold structure, enabling efficient and stable algorithms, including optimization (Oseledets, 2011). When such separability is present, FTTs provide accurate and efficiently evaluable representations.

Let $D = \times_{i=1}^{d}[a_i, b_i] \subset \mathbb{R}^d$ with $a_i < b_i$ for $i = 1, \dots, d$. A function $f \colon D \to \mathbb{R}$ is said to have FTT rank $\overline{r} = (\overline{r}_1, \dots, \overline{r}_{d-1}) \in \mathbb{N}^{d-1}$, if it can be written as

$$f(\boldsymbol{x}) = f(x_1, \dots, x_d) = F_1(x_1)F_2(x_2)\cdots F_d(x_d) \quad (11)$$

with matrix valued functions $F_i(x_i) \in \mathbb{R}^{\overline{r}_{i-1}, \overline{r}_i}$ for $i = 1, \dots, d$ with the convention $\overline{r}_0 = \overline{r}_d = 1$. Note that in the case $d = 2$, the FTT format (11) can be interpreted as a singular value decomposition (SVD) in function space, since $f(x_1, x_2) = F_1(x_1)F_2(x_2) = \sum_{k=1}^{\overline{r}_1}[F_1(x_1)]_k[F_2(x_2)]_k$. Thus, the FTT format can be viewed as a generalization of the SVD to high-dimensional functions, while retaining important properties such as closedness (Holtz et al., 2012b).

In order to make this format accessible for approximation, a discretization in each direction $x_i$ will be introduced. To this end, let $\mathcal{H} = \bigotimes_{i=1}^{d} \mathcal{H}_i(a_i, b_i)$ be a product Hilbert space on $D$, where each $\mathcal{H}_i(a_i, b_i)$ is equipped with a scalar product $(\cdot, \cdot)_{\mathcal{H}_i}$. For $\boldsymbol{m} = (m_1, \dots, m_d) \in \mathbb{N}^d$, using the notation in Table 1 in the appendix, define the discrete set

of $\mathcal{H}$-orthonormal tensor basis functions $\mathcal{B}_{\boldsymbol{m}} := \{\phi_{\boldsymbol{\alpha}} := \bigotimes_{i=1}^{d} \phi_{\alpha_i}^i \mid \boldsymbol{\alpha} \in [\boldsymbol{m}]\}$, with univariate $\mathcal{H}_i$-orthonormal basis functions $\phi_{\alpha_i}^i \colon [a_i, b_i] \to \mathbb{R}$. For $f$ with FTT rank $\overline{r}$, we may then approximate

$$f(\boldsymbol{x}) \approx \boldsymbol{C}[\Phi(\boldsymbol{x})] := \sum_{\boldsymbol{\alpha} \in [\boldsymbol{m}]} \boldsymbol{C}[\boldsymbol{\alpha}]\phi_{\boldsymbol{\alpha}}(x_1, \dots, x_d), \quad (12)$$

for $\Phi(\boldsymbol{x}) = (\phi_j^1(x_1))_{j=1}^{m_1} \otimes \cdots \otimes (\phi_j^d(x_d))_{j=1}^{m_d} \in \mathbb{R}^{\boldsymbol{m}}$ and a tensor array $\boldsymbol{C} \in \mathbb{R}^{\boldsymbol{m}}$ with (algebraic) tensor train (TT) rank $\boldsymbol{r} = (r_1, \dots, r_{d-1})^\top \in \mathbb{N}^{d-1}$ bounded by the FTT rank $\overline{r}$. In particular, we have the decomposition into a tensor train (or matrix product state) format as

$$\boldsymbol{C}[\boldsymbol{\alpha}] = \boldsymbol{C}_1[\alpha_1]\boldsymbol{C}_2[\alpha_2]\cdots\boldsymbol{C}_d[\alpha_d], \quad (13)$$

with order three tensors $\boldsymbol{C}_i \in \mathbb{R}^{r_{i-1}, m_i, r_i}$ defining matrices $\boldsymbol{C}_i[\alpha_i] = \boldsymbol{C}_i[:, \alpha_i, :] \in \mathbb{R}^{r_{i-1}, r_i}$ with the convention that $r_0 = r_d = 1$. The set of TTs of fixed rank $\boldsymbol{r}$ is denoted as $\mathcal{M}_{\boldsymbol{r}}^{\boldsymbol{m}}$. In particular, there exists a maximal possible TT rank $\overline{r} \in \mathbb{N}^{d-1}$ such that $\mathcal{M}_{\overline{r}}^{\boldsymbol{m}} = \mathbb{R}^{\boldsymbol{m}}$.

In what follows, we refer to a function in the form of the right hand side (12) as *extended tensor train* (xTT). The xTT format is a special case of the FTT format with a specific choice of univariate basis functions. The set of xTTs building on $\mathcal{M}_{\boldsymbol{r}}^{\boldsymbol{m}}$ is denoted as

$$\mathcal{X}_{\boldsymbol{r}}^{\boldsymbol{m}} := \left\{ \widehat{V}_{\boldsymbol{C}} = \boldsymbol{C}[\Phi(\cdot)] \mid \boldsymbol{C} \in \mathcal{M}_{\boldsymbol{r}}^{\boldsymbol{m}} \subset \mathbb{R}^{\boldsymbol{m}} \right\}. \quad (14)$$

Note that, by construction, we have $\mathcal{X}_{\boldsymbol{r}}^{\boldsymbol{m}} \subset \mathcal{H}$ for any rank $\boldsymbol{r} \leq \overline{r}$, and for any $\widehat{V}_{\boldsymbol{C}} \in \mathcal{X}_{\boldsymbol{r}}^{\boldsymbol{m}}$ it holds that $\|\widehat{V}_{\boldsymbol{C}}\|_{\mathcal{H}} = \|\boldsymbol{C}\|_{\mathrm{F}}$. Key properties of the considered nonlinear low-rank model class include storage complexity, multi-linearity and related efficient evaluation. Provided that the ranks can be bounded, the TT format exhibits a storage complexity based on the shape of each component tensor $\boldsymbol{C}_i$ with upper bound $\mathcal{O}(\max(m_1, \dots, m_d)d \max(r_1, \dots, r_{d-1})^2)$, which scales only linearly in the dimension $d$. Moreover, a function in xTT format can be evaluated fast and numerically stably, including possible access to gradients or Hessians, see Appendix A.4 for details. To this end, originated by (10), we are interested in solving a sequence of optimization tasks for $n = 1, \dots, N - 1$ given by

$$\min_{\widehat{V}_{\boldsymbol{C}} \in \mathcal{X}_{\boldsymbol{r}}^{\boldsymbol{m}}} \left\{ \widehat{\mathcal{L}}_n^K(\widehat{V}_{\boldsymbol{C}}) + \tau_n \|\widehat{V}_{\boldsymbol{C}}\|_{\mathcal{H}}^2 \right\} =$$
$$\min_{\boldsymbol{C} \in \mathcal{M}_{\boldsymbol{r}}^{\boldsymbol{m}}} \left\{ \widehat{\mathcal{L}}_n^K(\widehat{V}_{\boldsymbol{C}}) + \tau_n \|\boldsymbol{C}\|_{\mathrm{F}}^2 \right\} \quad (15)$$

for some regularization magnitude $\tau_n > 0$. Note that the regularization term involves the $\mathcal{H}$ norm, see Appendix A.3.1. Recall that we are interested in approximating the true value function $V(\mathbf{x}, \mathbf{t})$, which is expected to have varying $\mathcal{H}$ norm over time snapshots $t = t_0, \dots, t_N$. Consequently, to relate

the regularized loss from (15) to the original loss from (10), the regularization magnitude needs to be chosen suitably and adaptively, which we explain in detail in Appendix A.6.1. Further, it is crucial to choose the TT rank $r$ and the numbers of degrees of freedom $m$ appropriately, and we provide a corresponding adaptive strategy in Appendix A.6.2 and Appendix A.6.3. Moreover, a detailed discussion of choices for tensor basis functions, along with their associated advantages and challenges, is given in Appendix A.3. Finally, a discussion of the FTT rank analysis for certain classes of functions is provided in Appendix A.7.

## 3.1 Optimization via alternating least squares

The set $\mathcal{M}_r$ defines a Riemannian manifold (Holtz et al., 2012b), rendering the option for Riemannian optimization. Here, we consider an alternative approach for minimization based on alternating least squares (ALS), cf. Holtz et al. (2012a). In particular, instead of solving (15) via iterates of the whole TT $C$, we split the minimization into iterations of so-called *sweeps* consisting of several *micro steps*. In each micro step we update the $i$-th component of the TT only, while fixing all other components. For this, we compactly write (see Appendix A.1)

$$C = C_1 C_2 \cdots C_d = U_1 \cdots U_{i-1} \mathfrak{C} U_{i+1} \cdots U_d$$

with orthogonal order three tensors $U_j \in \mathbb{R}^{r_{j-1}, m_j, r_j}$ for $j = 1, \ldots, i-1, i+1, \ldots d$ and a so-called *core* $\mathfrak{C} \in \mathbb{R}^{r_{i-1}, m_i, r_i}$, see Appendix A.2 for more details. Then, at each micro step, the loss from (15) reduces to the problem

$$\min_{\mathfrak{C}} \|A_{i,n}^u \operatorname{vec}(\mathfrak{C}) - y_{n+1}\|_2^2 + \tau_n \|\mathfrak{C}\|_F^2, \qquad (16)$$

with a matrix $A_{i,n}^u \in \mathbb{R}^{K, r_{i-1} m_i r_i}$ and $y_{n+1} = (y_{n+1}^{(k)})_k \in \mathbb{R}^K$, see Appendix A.5 for a derivation.

After solving the minimization task stated in (15), which yields a TT $C^n \in \mathcal{M}_r^m$, we then obtain an approximation of our value function $V(\cdot, t_n)$ at time $t = t_n$, denoted by $\widehat{V}_{C^n} \in \mathcal{X}_r^m$, in xTT format, i.e.

$$V(\boldsymbol{x}, t_n) \approx \widehat{V}_n(\boldsymbol{x}) \approx \widehat{V}_{C^n} = C^n[\Phi(\boldsymbol{x})], \ \forall \boldsymbol{x} \in D. \quad (17)$$

*Remark* 3.1 (Extension strategies for the domain). The approximation of the value function $V$ is only defined locally on a domain $D \Subset \mathbb{R}^d$. In practice, this domain is given by $D = D_n^u$, which depends on both the time step and the current policy $u$. If, during the sampling dynamics in (3), the required policy evaluation $u^* = -\sigma^\top \nabla V$ involves points outside of $D$, suitable extension strategies must be employed. We discuss these strategies in detail in Appendix A.4.4.

*Remark* 3.2 (Initialization of xTTs). The minimization requires initial xTTs. At the final time $t = t_N$, we initialize

via an $L^2$ regression of the log-target density using samples from the initial sampling dynamics. Given an approximate xTT at $t_{n+1}$, the initialization at $t_n$ is obtained by a generalized Galerkin-type projection of this xTT, accounting for the domain change; see Appendix A.4.5.

*Remark* 3.3 (Stabilization of backward regressions). While the backward iteration in Algorithm 1 combined with FTTs offers clear advantages over iterative loss minimization with neural networks, it also poses stability challenges. Addressing these issues is a key contribution of this work; the corresponding stabilization strategies are detailed in the appendix. These include the choice of suitable basis functions (Appendix A.3), handling moving domains along sample trajectories and evaluations outside these domains (Appendices A.4.4 and A.4.5), adaptive regularization (Appendix A.6.1), adaptive rank selection (Appendix A.6.2), and adaptive basis degree selection (Appendix A.6.3).

# 4 Numerical experiments

In this section we evaluate our sampling algorithm on challenging high-dimensional problems. We refer to an additional problem from computational statistics in Appendix B.5.

## 4.1 Highdimensional Multiwell problems

We first consider a class of Multiwell problems, defined by

$$\rho_{\text{target}}(\boldsymbol{x}) = \exp\left(-\sum_{i=1}^{m}(x_i^2 - \delta)^2 - \frac{1}{2}\sum_{i=m+1}^{d} x_i^2\right), \ (18)$$

which are commonly used benchmarks (Berner et al., 2024; Richter & Berner, 2024; Wu et al., 2020; Midgley et al., 2022). These densities are chosen because they resemble physics and molecular dynamics problems, providing an indication of performance in practical applications. At the same time, they offer a controlled setting for systematically studying the three key challenges in sampling: (1) the dimensionality $d$, (2) the mode separation $\delta$, and (3) the number of modes $2^m$. In our experiments, we use a Fourier basis (see Appendix A.3.1) with 3 to 13 basis functions per dimension, treating this as a hyperparameter tuned according to the observed approximation quality. We employ an $H_{\text{mix}}^2$ orthonormalization as described in Appendix A.3.2. For the noising process (2), we set $f(\boldsymbol{x}, t) = \boldsymbol{x}$, $\sigma(t) = \sqrt{2}$, and consider a time horizon $T = 2$. We apply Algorithm 1 to two Multiwell problems, with dimensions $d = 10$ and $d = 50$ and 8 and 32 modes, respectively, with $\delta = 2$.

As shown in Figure 3, increasing the number of time steps $N$ (which also determines the number of functions $\widehat{V}_n$ approximated) steadily improves sampling performance, ultimately achieving remarkable quality. Importantly, the algorithm's runtime is significantly faster than that of alternative, pre-

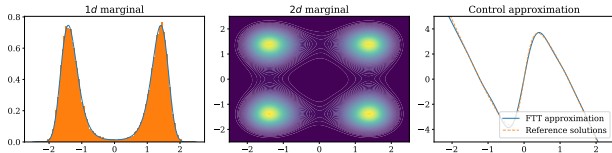

*Figure 2.* We plot one- and two-dimensional marginals of our Multiwell problem in $d = 50$, showing very high sampling accuracy that aligns with the metrics in Figure 3. Notably, our algorithm does not exhibit mode collapse, a common challenge in diffusion-based sampling (cf. Figure 1). On the right-hand side, we compare the first component of the learned control with a reference solution (computed via finite differences on one-dimensional problems) at $t = 1.9$ for $x$-values varying along a single dimension.

dominantly neural network-based samplers. In $d = 10$, training ranges roughly from 1 minute ($N = 2^8$) to 20 minutes ($N = 2^{13}$), while in $d = 50$ it ranges from 10 to 300 minutes. Note that the runtime grows linearly with $N$. We refer to Figure 4 for a comparison with the *time-reversed diffusion sampler* (DIS) from (Berner et al., 2024), which considers the same noising SDE, however relies on neural network approximations and variational training. We refer to Figure 14 in Appendix B.4 for a comparison to further sampling methods, showing significantly improved performance. Finally, we refer to Figure 2 comparing marginals of the 50 dimensional problem with a reference solution.

*Remark* 4.1 (Approximation of $\log Z$ and the PDE solution). We note that the Girsanov theorem relates the error in the log-normalizing constant $\log \mathcal{Z}$ to the $L^2$-approximation of the ground-truth score $\nabla \log p$, i.e., the gradient of the solution to the HJB equation w.r.t. to the measure induced by $p_{\text{target}}$ (Berner et al., 2024). Under suitable assumptions, one can further leverage Poincaré inequalities to relate this error to the $L^2$-approximation of the PDE solution itself, measuring the accuracy of our proposed PDE solver.

### 4.2 Investigation of optimal ranks

In this section, we study highly anisotropic Gaussian targets to analyze the rank structure of the induced transport problems and to assess the adaptivity of our algorithm. We consider target densities with potential $\log \rho_{\text{target}}(\boldsymbol{x}) = -\boldsymbol{x}^\top M \boldsymbol{x}$, where $M$ is a random full-rank matrix. For this class, the maximal admissible rank is $\mathbf{r} = (3, 4, \ldots, 2 + d/2, \ldots, 4, 3) \in \mathbb{N}^{d-1}$ for even $d$; see Corollary A.6 for the general

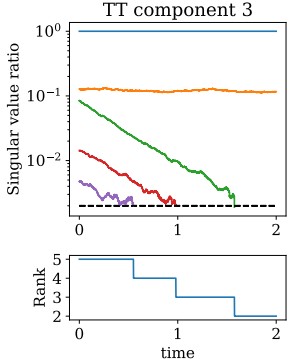

*Figure 5.* We display the singular values of TT component 3 over time and reduce the rank whenever a corresponding value is below a prespecified threshold.

case. As the target distribution converges toward the standard normal, $p_{Y_T} \approx p_{\text{prior}} = \mathcal{N}(0, \mathrm{Id})$, the ranks are expected to approach the constant vector $\mathbf{r} = (2, \ldots, 2)$. This behavior is confirmed in Figure 5, which shows the evolution of singular values and the corresponding rank adaptivity, resulting in a reduced runtime. The clear spectral gaps demonstrate that the algorithm reliably identifies and adapts to the appropriate ranks over time; see also Figure 11 in the appendix.

### 4.3 $\phi^4$ scalar field theory

Next, we evaluate our method on the discretized Ginzburg-Landau model (specifically, the $\phi^4$ scalar field theory), a canonical framework in statistical mechanics used to describe continuous phase transitions and spontaneous symmetry breaking (Ginzburg & Landau, 1950; Rosenstein & Li, 2021). The target density is defined by a Gibbs distribution over a lattice, characterized by a highly non-convex energy landscape resulting from the competing forces of local bistability and nearest-neighbor spatial coupling,

$$\rho_{\text{target}}(\boldsymbol{x}) = \exp\left( -\frac{1}{2} \sum_{i=1}^{d-1} (x_i - x_{i+1})^2 \right.$$
$$\left. -\frac{1}{2} \sum_{i=1}^{m} (x_i^2 - \delta)^2 - \frac{1}{2} \sum_{i=m+1}^{d} x_i^2 \right). \tag{19}$$

This system is highly challenging for sampling algorithms, as the target distribution exhibits exponentially many metastable modes separated by high-energy barriers, leading to severe mode collapse in standard MCMC methods. Nearest-neighbor interactions further induce strong dependencies between lattice sites, resulting in a highly constrained and correlated high-dimensional geometry. For $\delta = 2$, Figure 6 shows that the TTD method generates high-quality samples in two distinct multimodal regimes.

## 5 Conclusion & Discussion

In this work, we introduced Tensor Train Diffusion (TTD), an efficient PDE solver for high-dimensional HJB equations leading to a novel way of sampling from unnormalized densities. Our method solves the HJB equation underlying the diffusion process using a functional tensor train (FTT) representation. By integrating this with a backward-in-time iterative scheme derived from BSDEs, TTD provides a fast, accurate, and stable alternative to neural network techniques. In particular, TTD requires less target evaluations and does not rely on SGD-based optimization with long training times and hyperparameter sensitivity.

In general, our TTD framework presents a solid foundation for the robust and efficient solution of HJB equations, that can readily be used for different sampling problems – we

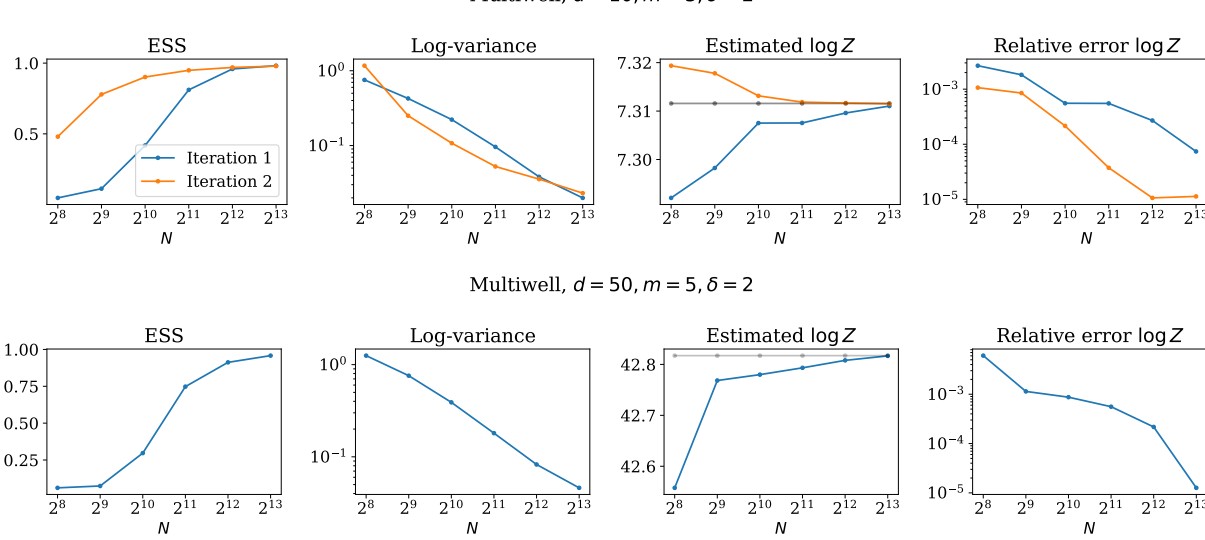

*Figure 3.* We consider two instances of the Multiwell problem defined in (18) and evaluate performance using the effective sample size (ESS), log-variance divergence, as well as the log-normalizing constant and its relative error; see Appendix B.2. As expected, increasing the number of steps $N$ leads to improved performance and the outer loop can provide additional gains. Remarkably, both the ESS and relative error remain stable even in the more challenging $d = 50$ case.

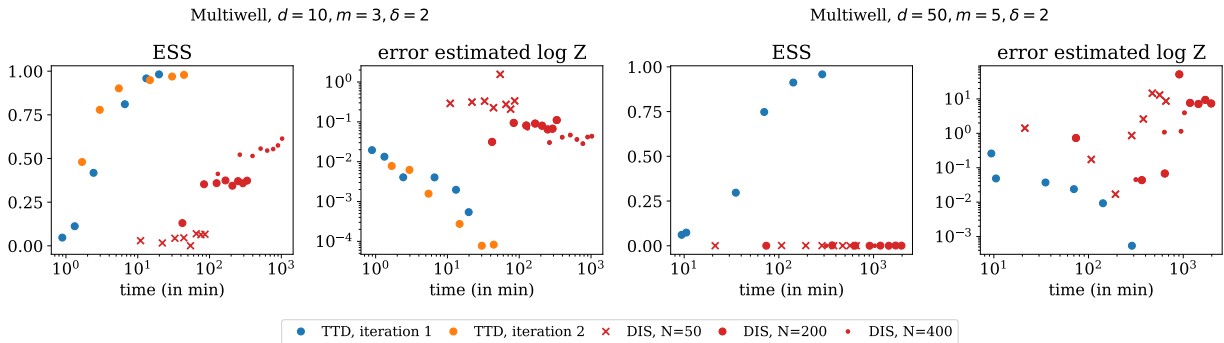

*Figure 4.* We compare the performance versus runtime of our TTD sampler with DIS (Berner et al., 2024). By design, our algorithm produces one result per chosen number of steps $N$ (shown as blue and orange dots), whereas DIS can improve over training time. Accordingly, we evaluate DIS at equally spaced runtime intervals. In both experiments, our algorithm is not only significantly faster but also achieves better results, particularly in settings where DIS can exhibit instability.

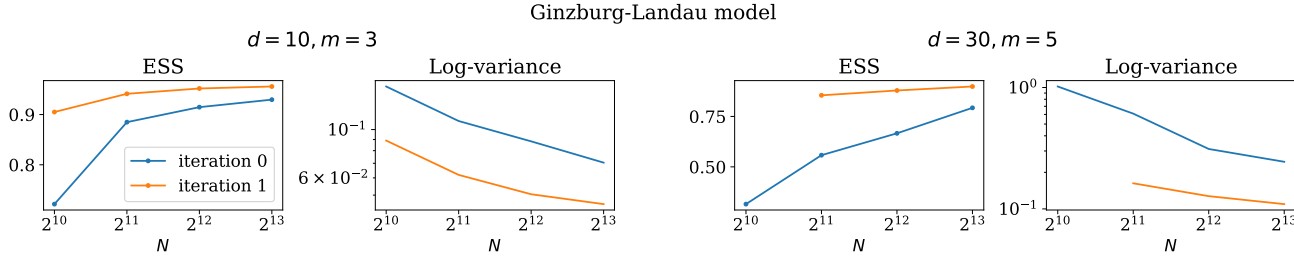

*Figure 6.* We consider two instances of the Ginzburg-Landau model defined in (19) and assess performance using the effective sample size (ESS) and the log-variance divergence. In both settings, we obtain strong results, with performance improving as the number of steps $N$ and the number of outer iterations increase.

refer to potential limitations in Appendix B.1. While we provide a significantly accelerated tensor train implementation in PyTorch, we believe that further hardware-specific optimization is possible. On the theoretical side, our work motivates further research to analyze latent low-rank structures of different target densities for educated choices of basis functions and variable orderings. Future work could also combine the framework with neural network models or classical sampling methods, as well as extend TTD to broader classes of PDEs and applications. From a theoretical perspective, an open question concerns the behavior of FTT ranks for solutions of the underlying HJB equation, which is only partially understood (Gruhlke et al., 2026). In the general case, the current state-of-the-art theory (Ali & Nouy, 2020a;b; 2021; Griebel et al., 2023; Bachmayr, 2023) does not directly apply and would need to be extended to weighted regularity classes, since solutions of HJB equations are defined on $\mathbb{R}^d$ and are generally unbounded as $\|\boldsymbol{x}\| \to \infty$. Consequently, on the computational side, one must rely on rank-adaptive approaches, i.e., determining the xTT ranks for the approximation in an adaptive manner.

## Impact Statement

This paper presents work whose goal is to advance the field of Machine Learning. There are many potential societal consequences of our work, none which we feel must be specifically highlighted here.

## Acknowledgements

The authors thank Denis Blessing for many useful discussions. The research of L.R. was partially funded by Deutsche Forschungsgemeinschaft (DFG) through the grant CRC 1114 "Scaling Cascades in Complex Systems" (project A05, project number 235221301). The research of R.G. was funded by the Deutsche Forschungsgemeinschaft (DFG, German Research Foundation) under Germany's Excellence Strategy (EXC-2046/2, project ID 390685689).

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

# Appendix

## Appendix Contents

## A  Extended tensor trains

This section will give a deeper understanding in the concepts of tensor trains and extended tensor trains. We note that the concept of an extended tensor train in dimension $d = 1$ reduces to the approximation a linear vector space of basis functions. For $d = 2$ the concept of tensor trains coincides with singular value decomposition. Since singular value decomposition in higher dimension, i.e. the decomposition of a tensor into sums of rank-1 tensors is not a closed format, suitable for well-defined optimization tasks, we utilize the framework of tensor trains, which form a closed Riemannian manifold in $\mathbb{R}^{\boldsymbol{m}}$ (Holtz et al., 2012b).

### A.1  Contraction of tensors

Let $\boldsymbol{m} \in \mathbb{N}^d$ be the *mode sizes*. For a tensor $\boldsymbol{A} \in \mathbb{R}^{\boldsymbol{m}}$, we refer to the tensor slice in the $k$-th dimension as the $k$-th mode dimension. A tensor with only one mode dimension is a vector, a tensor with only two mode dimensions is a matrix, a tensor with three mode dimensions is an order three tensor, etc.

First we will define the key concept of *contraction* between tensors. Let $k, \ell \in \mathbb{N}$ with $\ell > k$. For any tensor $\boldsymbol{A} \in \mathbb{R}^{m_1, \dots, m_k}$ and $\boldsymbol{B} \in \mathbb{R}^{m_k, \dots, m_\ell}$, we define $\boldsymbol{AB} \in \mathbb{R}^{m_1, \dots, m_{k-1}, m_{k+1}, \dots, m_\ell}$ as the tensor resulting from the contracting of neighbored

| | |
|---|---|
| $\boldsymbol{m} \in \mathbb{N}^d$ | dimension array $\boldsymbol{m} = (m_1, \ldots, m_d)$ |
| $k\boldsymbol{m} + l$ | $(km_1 + l, \ldots, km_d + l)$ for $k, l \in \mathbb{N}_0$ |
| $[\boldsymbol{m}]$ | indexing $[\boldsymbol{m}] = \times_{i=1}^{d}\{1, \ldots, m_i\}$ |
| $\boldsymbol{m}_1 \geq \boldsymbol{m}_2, \boldsymbol{m} \geq k$ | component wise comparison $\boldsymbol{m}, \boldsymbol{m}_1, \boldsymbol{m}_2 \in \mathbb{N}^d, k \in \mathbb{N}$ |
| $\boldsymbol{\alpha}, \boldsymbol{\beta}, \boldsymbol{\gamma}$ | multiindex in $\mathbb{N}^d$ |
| $\mathbb{R}^{\boldsymbol{m}}$ | tensor space $\mathbb{R}^{m_1, \ldots, m_d}$ |
| $\boldsymbol{A}, \boldsymbol{B}, \boldsymbol{C}$ | tensor elements in $\mathbb{R}^{\boldsymbol{m}}$ |
| $\boldsymbol{r}$ | rank $\boldsymbol{r} = (r_1, \ldots, r_{d-1})$ in $\mathbb{N}^{d-1}$ |
| $\boldsymbol{r}^1 \boldsymbol{r}^2$ | multiplication $\boldsymbol{r}^1 \boldsymbol{r}^2 = (r_1^1 r_1^2, \ldots, r_{d-1}^1 r_{d-1}^2)$ in $\mathbb{N}^{d-1}$ |
| $k_i, l_i$ | rank enumeration indices in $\{1, \ldots, r_i\}$ |
| $A_i, B_i, C_i$ | component order 3 tensor in $\mathbb{R}^{r_{i-1}, m_i, r_i}$ with entries indexed by $[k_{i-1}, \alpha_i, k_i]$ |
| $A_i[\alpha_i]$ | matrix extraction $A_i[\alpha_i] = A_i[:, \alpha_i, :] \in \mathbb{R}^{r_{i-1}, r_i}$ of component tensor $A_i$ |
| $A_i[k_{i-1}, :, k_i]$ | vector extraction in $\mathbb{R}^{m_i+1}$ for each rank enumeration $k_{i-1}, k_i$ |
| $\boldsymbol{A}[\boldsymbol{\alpha}]$ | tensor indexing $\boldsymbol{A}[\alpha_1, \ldots, \alpha_d]$ for $\boldsymbol{A} \in \mathbb{R}^{\boldsymbol{m}}, \boldsymbol{\alpha} \in [\boldsymbol{m}], \boldsymbol{m} \in \mathbb{N}^d$ |
| $\|\cdot\|_{\mathrm{F}}$ | Frobenius norm |

*Table 1.* List of compact notations used in this work.

mode dimensions, i.e. the last mode dimension of $\boldsymbol{A}$ and the first of $\boldsymbol{B}$:

$$\boldsymbol{A}\boldsymbol{B} = \sum_{i_k=1}^{m_k} \boldsymbol{A}[\ldots, i_k]\boldsymbol{B}[i_k, \ldots],$$

where we used Python type notation for easier understanding.

More generally, for two tensors $\boldsymbol{C} \in \mathbb{R}^{\boldsymbol{m}^1}$ and $\boldsymbol{D} \in \mathbb{R}^{\boldsymbol{m}^2}$ for $\boldsymbol{m}^1 \in \mathbb{N}^{d_1}$ with $\boldsymbol{m}^1 = (m_1^1, \ldots, m_{d_1}^1)$ and $\boldsymbol{m}^2 \in \mathbb{N}^{d_2}$ with $\boldsymbol{m} = (m_1^2, \ldots, m_{d_2}^2)$ such that the $k$-th and the $\ell$-th mode size $m_k^1$ and $m_\ell^2$ coincide, i.e. $m = m_k^1 = m_\ell^2$, we define the contraction with respect to the $k$-th and $\ell$-th mode direction as the tensor $\boldsymbol{E} \in \mathbb{R}^{m_1^1, \ldots, m_{k-1}^1, m_{k+1}^1 m_{d_1}^1, m_1^2, \ldots, m_{\ell-1}^2, m_{\ell+1}^2, m_{d_2}^2}$ with

$$\boldsymbol{E} = \boldsymbol{C} \circ_{kl} \boldsymbol{D} = \sum_{i=1}^{m} \underbrace{\boldsymbol{C}[\ldots, i, \ldots]}_{k\text{-th slice}} \underbrace{\boldsymbol{D}[\ldots, i, \ldots]}_{\ell\text{-th slice}}.$$

In particular, using this notation, it holds that

$$\boldsymbol{A}\boldsymbol{B} = \boldsymbol{A} \circ_{k,1} \boldsymbol{B}.$$

## A.2 Degrees of freedom, the core and remaining orthogonal components

This section is devoted to representations of tensor trains (TTs) suitable for numerically stable usage, in particular for the high-dimensional case.

Consider a TT with rank $\boldsymbol{r} = (r_1, \ldots, r_{d-1}) \in \mathbb{N}^{d-1}$ given as

$$\boldsymbol{C}[\boldsymbol{\alpha}] = C_1[\alpha_1]C_2[\alpha_2]\cdots C_d[\alpha_d], \qquad \forall \boldsymbol{\alpha} \in [\boldsymbol{m}] \tag{20}$$

with $\boldsymbol{C}_i \in \mathbb{R}^{r_{i-1}, m_i, r_i}$ with the convention of $r_0 = r_d = 1$.

Using the notation of contractions of tensors from Appendix A.1, we can compactly write

$$\boldsymbol{C} = C_1 C_2 \cdots C_d. \tag{21}$$

First, we note that for any $i = 1, \ldots, d-1$ we can choose an arbitrary invertible matrix $G_i \in \mathrm{GL}(r_i) \subset \mathbb{R}^{r_i, r_i}$ and insert it and its inverse between the $i$-th and $(i+1)$-th component without changing the represented full tensor. In particular, let

$$\widehat{C}_i = C_i G_i, \qquad \widehat{C}_{i+1} = G_i^{-1} C_{i+1}. \tag{22}$$

Then,

$$C = C_1 C_2 \cdots C_d = C_1 \cdots \widehat{C}_i \widehat{C}_{i+1} \cdots C_d. \tag{23}$$

The space $\mathrm{GL}(r_i)$ is of dimension $r_i^2$ and, consequently, the degrees of freedoms in a TT are given by

$$\#\mathrm{d.o.f.}(\mathrm{TT}) = \underbrace{\sum_{i=1}^{d} r_{i-1} m_i r_i}_{\text{raw number of entries for each component}} - \underbrace{\sum_{i=1}^{d-1} r_i^2}_{\text{dimensions of } \mathrm{GL}(r_i)}. \tag{24}$$

For further readings regarding the derivation of degrees of freedoms in terms of gauge conditions, we refer to Oseledets (2011); Holtz et al. (2012b); Uschmajew & Vandereycken (2013).

Now, we can apply high-order singular value decomposition (HOSVD, see Grasedyck (2010); Oseledets (2011)) on $C$. Let us fix $1 \leq k \leq d$ as the so-called *core position*. Then for any $i = 1, \ldots, k-1$, we iteratively reshape the $i$-th TT component $C_i \in \mathbb{R}^{r_{i-1}, m_i, r_i}$ as a matrix $C_i \in \mathbb{R}^{r_{i-1}, m_i r_i}$ apply a singular value decomposition

$$C_i = U_i \Sigma_i V_i^\top, \tag{25}$$

and contract the non-orthonormal part $\Sigma_i V_i^\top$ from left to $C_{i+1}$ and redefine

$$C_{i+1} \leftarrow \Sigma_i V_i^\top C_{i+1}. \tag{26}$$

Analogously, for $i = d, d-1, \ldots, k+1$ we iteratively perform SVDs $C_i = U_i \Sigma_i V_i^\top$ and contract $U_i \Sigma_i$ from right to $C_{i-1}$. Finally, we can reshape each matrix $U_i \in \mathbb{R}^{r_{i-1}, m_i r_i}$ to an order three tensor $U_i \in \mathbb{R}^{r_{i-1}, m_i, r_i}$ for $i = 1, \ldots, k-1$ and matrix $V_i^\top \in \mathbb{R}^{r_{i-1}, m_i r_i}$ to an order three tensor $U_i \in \mathbb{R}^{r_{i-1}, m_i, r_i}$ for $i = d, d-1, \ldots, k+1$. Then, it holds that

$$C = U_1 \cdots U_{k-1} C_k U_{k+1} \cdots U_d. \tag{27}$$

The resulting updated non-orthonormal component $\mathfrak{C} = C_k$ is called the *core* at position $k$ of $C$.

The classical representation of a TT from (20) is prone to rounding errors, when trying to access $C[\boldsymbol{\alpha}]$ with $\boldsymbol{\alpha} = (\alpha_1, \ldots, \alpha_d)$. Instead, one first defines a core representation with core $\mathfrak{C}$, e.g. with a core position $k$. Then, the orthogonal components are contracted first and in the end contracted with the core, leading to a numerically more stable result. In particular, a left contraction computationally realized from left left to right given as

$$L_k[\alpha_1, \ldots, \alpha_{k-1}] = U_1[\alpha_1] \cdots U_{k-1}[\alpha_{k-1}], \qquad L_k := 1 \text{ for } k = 1, \tag{28}$$

and a right contraction $R_k$ computationally realized from right to left contractions is defined as

$$R_k[\alpha_k, \ldots, \alpha_d] = U_{k+1}[\alpha_{k+1}] \cdots U_d[\alpha_d], \qquad R_k = 1 \text{ for } k = d. \tag{29}$$

Finally, the contraction reads

$$C = L_k \mathfrak{C} R_k. \tag{30}$$

### A.3 Tensor basis functions

We emphasize that the basis functions are required to have *local support*. This requirement is motivated by two structural properties arising from the approximation of the value function. First, the value function is, up to an additive constant, a negative log-density with unbounded support on $\mathbb{R}^d$. Functions of this type generally do not belong to classical smoothness spaces defined via global integrability, such as standard Sobolev spaces. Second, the associated density typically concentrates most of its mass in localized regions of the domain. These regions are precisely those relevant for accurate approximation, which naturally motivates a localized representation. In our algorithm, the value function is approximated along sample trajectories. In an idealized setting, for a value function $V$ with density proportional to $e^{-V}$, and assuming exact reverse sampling dynamics at time $t$, one would aim to approximate $V(\cdot, t)$ in spaces such as $L^2(\mathbb{R}^d, e^{-V(\cdot, t)} \, \mathrm{d}\mathbf{x})$, or more generally in weighted Sobolev spaces. From this perspective, restricting the approximation to compact subdomains $K$ can be interpreted as a localization or truncation of these weighted function spaces.

### A.3.1 ORTHONORMAL UNIVARIATE BASIS FUNCTIONS

Let $a_i < b_i$ and let $\mathcal{H}_i(a_i, b_i)$ be a Hilbert space with inner product $(\cdot, \cdot)_{\mathcal{H}_i}$. We introduce the corresponding basis functions and discuss the limitations and challenges arising from approximation accuracy, localization capability, and robustness with respect to sampling. We then present the concept of orthonormalization. Note that sample robustness is independent of the particular representation of the basis, including whether it is orthonormalized, and depends only on the vector space spanned by the basis functions.

**Legendre polynomials.** As a classical basis, we consider the Legendre polynomials $L_0, L_1, L_2, \ldots$.

- *Approximation*: Legendre polynomials exhibit asymptotic spectral convergence for analytic target functions.

- *Localization*: Functions with localized features may require a high polynomial degree for accurate approximation; in practice, the pre-asymptotic constant dominates and slows convergence.

- *Sample robustness*: For empirical $L^2$ regression problems, a Legendre basis requires a number of samples that grows polynomially with the number of degrees of freedom in order to guarantee, with high probability, a unique least-squares solution; see Cohen & Migliorati (2017). This restricts the feasible number of basis functions when only limited samples are available.

**Spline polynomials of smoothness $s$ and order $p$.** As a second ansatz, we consider spline functions with local smoothness $s$ and local polynomial degree $p$, defined on a grid $a_i = x_0 < x_1 < \ldots < x_k = b_i$, with $k + 1$ knots and constructed via the Cox-de Boor recursion formulas; see De Boor & De Boor (1978).

- *Approximation*: Spline functions achieve algebraic convergence rates in $p$, depending on the grid width; see Schumaker (2007).

- *Localization*: Localized features can be captured much more effectively than with global polynomial bases.

- *Sample robustness*: Empirical regression requires samples that adequately cover the domain. Due to the local support of the basis, if only few samples fall between neighboring knots, the reconstruction may fail even for simple target functions; see Appendix A.3.1.

Note, that also rank regularization is not enough to correct for the stable regression using splines under various sample distributions. This drawback becomes in particular critical in cases of our algorithm, when the initial policy does not allow for proper mass distribution, e.g. for multimodal targets.

**Fourier modes.** Fourier modes of the form $1$, $\sin(k\omega x)$, and $\cos(k\omega x)$ for $k = 1, 2, \ldots$ constitute a classical basis class.

- *Approximation*: Fourier modes exhibit spectral approximation rates for analytic functions that admit a periodic extension of all derivatives. For non-periodic functions, however, the convergence deteriorates to algebraic rates, typically of low order, depending on boundary regularity.

- *Localization*: As with Legendre polynomials, localized features generally require a large number of Fourier modes for accurate representation. Moreover, non-periodic target functions may induce spurious oscillations (Gibbs-type phenomena) in the approximation.

- *Sample robustness*: Fourier bases are known to exhibit favorable sample complexity and robustness properties, in particular compared to Legendre polynomials; see Trunschke (2022). This allows for the use of a large number of basis functions without a corresponding explosion in the required number of samples for stable regression.

**Extended Fourier modes.** This basis augments Fourier modes with low-order Legendre polynomials $L_0 \equiv 1, L_1, L_2, \ldots$ together with $\sin(k\omega x)$ and $\cos(k\omega x)$ for $k = 1, 2, \ldots$.

- *Approximation*: The inclusion of polynomial modes up to degree $p$ mitigates the loss of convergence caused by non-periodicity. In particular, for an enrichment up to degree $p = 2$, the approximation of smooth non-periodic functions achieves algebraic convergence of order $p + 1 = 3$.

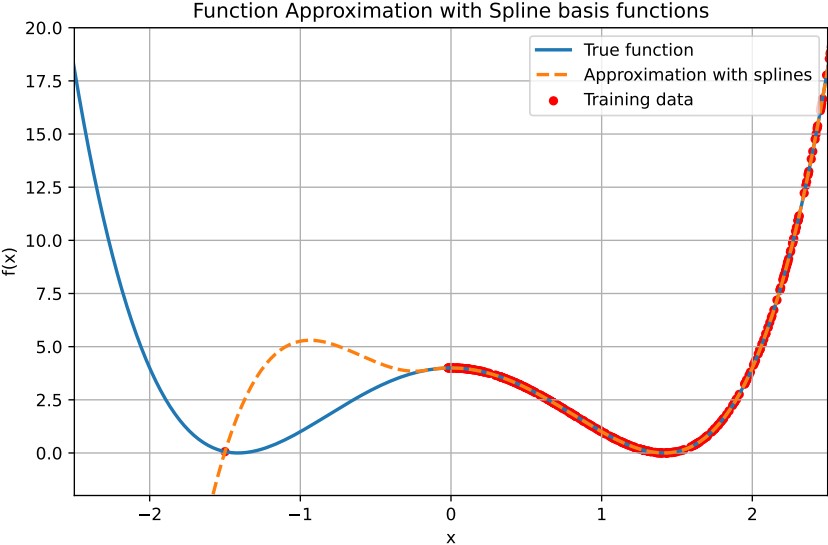

*Figure 7.* Consider a spline basis with 3 knots, local polynomial degree 4, and smoothness parameter $s = 1$, used to approximate a target function from a limited number of samples concentrated on the left side of the origin. Despite the weak coupling induced by the smoothness constraint $s$, the resulting regression based on these samples performs very poorly.

- *Localization*: The localization properties are comparable to those of the underlying Legendre and Fourier components.

- *Sample robustness*: This construction combines advantages from both Legendre and Fourier bases. In particular, it is expected to improve sample complexity and stability relative to pure polynomial bases, thereby enabling a larger number of basis functions with controlled sample requirements. While a rigorous analysis is beyond the scope of this work, empirical results support this behavior.

### A.3.2 ORTHONORMAL TENSOR BASIS FUNCTIONS AND CHOICE OF $\mathcal{H}$

For $i = 1, \ldots, d$, let $\Phi_{i,\text{raw}} = (\phi_{\alpha_i,\text{raw}})_{\alpha_i=1}^{m_i}$ denote the stacked basis vector of $m_i \in \mathbb{N}$ raw univariate basis functions, e.g. Legendre polynomials, Splines, extended Fourier modes. Then we define the associated Gramian matrix $G_i \in \text{GL}(m_i)$ with entries defined as

$$[G_i]_{k,\ell} = (\phi_{\alpha_k,\text{raw}}, \phi_{\alpha_\ell,\text{raw}})_{\mathcal{H}_i(a_i,b_i)}, \quad k, \ell = 1, \ldots, m_i. \tag{31}$$

Then

$$(\phi_j^i)_{j=1}^{m_i} := G_i^{-1/2} \Phi_{i,\text{raw}} \tag{32}$$

defines a vector of $\mathcal{H}_i(a_i, b_i)$ orthonormal basis functions.

Then, tensorization of these, yields the basis set

$$\mathcal{B}_{\boldsymbol{m}} := \left\{ \phi_{\boldsymbol{\alpha}} := \bigotimes_{i=1}^{d} \phi_{\alpha_i}^i \ \middle| \ \boldsymbol{\alpha} \in [\boldsymbol{m}], (\phi_{\alpha_j}^i, \phi_{\alpha_k}^i)_{\mathcal{H}_i} = \delta_{jk}, \forall i, j, k \right\}, \tag{33}$$

which forms an $\mathcal{H}$-orthonormal basis for the product Hilbert space $\mathcal{H} = \bigotimes_{i=1}^{d} \mathcal{H}_i(a_i, b_i)$.

In this work, we set $\mathcal{H}_i(a_i, b_i) = H^2(a_i, b_i)$, the Sobolev space of order (smoothness) 2. It holds for $K = \bigtimes[a_i, b_i]$ that

$$\mathcal{H} = H^2_{\text{mix}}(K), \tag{34}$$

where $H^s_{\text{mix}}(K)$ denotes the classical space of mixed regularity of smoothness $s$.

This choice of Hilbert space is motivated by two considerations. First, the BSDE loss in (9) requires integrability of both zero- and first-order derivatives. Second, to mitigate potential oscillatory behavior arising in the approximation step, we

additionally regularize second-order derivatives. This provides control over oscillations in the first derivative and thereby yields a more stable approximate policy.

The choice of orthonormalization is motivated by the use of the Parseval identity within the optimization scheme. Indeed, for $\widehat{V}_C = C[\Phi(\cdot)] \in \operatorname{span} \mathcal{B}_{\boldsymbol{m}}$, the identity $\|\widehat{V}_C\|_{\mathcal{H}} = \|C\|_{\mathrm{F}}$ is preserved at the level of the micro-steps of the alternating minimization scheme. This provides a direct interpretation of the regularization term $\tau_n\|\mathfrak{C}\|_{\mathrm{F}}$ in (16). In particular, for a tensor train (TT) coefficient tensor $C$ with core representation as in (27), and corresponding core tensor $\mathfrak{C}$, it holds that

$$\|C\|_{\mathrm{F}} = \|\mathfrak{C}\|_{\mathrm{F}}. \tag{35}$$

Consequently, we obtain $\|\mathfrak{C}\|_{\mathrm{F}} = \|\widehat{V}_C\|_{\mathcal{H}}$, i.e., the Frobenius norm of the core tensor coincides with the $\mathcal{H}$-norm of the represented function $\widehat{V}_C$.

## A.4 Evaluation, gradient and Hessian of extended tensor trains

This section is devoted to illustrating the efficient and numerically stable evaluation of functions in xTT format, including their gradients and Hessians.

To this end, let $\widehat{V}_C \in \mathcal{X}_{\boldsymbol{r}}^{\boldsymbol{m}}$ be an extended tensor train as defined in (14). Then, using $k_0 = k_d = 1$, it holds

$$\widehat{V}_C(\boldsymbol{x}) = C[\Phi(\boldsymbol{x})] = \sum_{\boldsymbol{\alpha} \in [\boldsymbol{m}]} C[\boldsymbol{\alpha}]\phi_{\boldsymbol{\alpha}}(x_1, \ldots, x_d) \tag{36a}$$

$$= \sum_{k_1=1}^{r_1} \cdots \sum_{k_{d-1}=1}^{r_{d-1}} C_1[k_0, \alpha_1, k_1] \cdots C_1[k_{d-1}, \alpha_d, k_d] \prod_{i=1}^{d} \phi_{\alpha_i}^i(x_i). \tag{36b}$$

### A.4.1 EVALUATION

Now define the vector $w^i = w^i(x_i)$ as

$$[w^i(x_i)]_j = (\phi_j^i(x_i))_{j=1}^{m_i} \in \mathbb{R}^{m_i}. \tag{37}$$

Then,

$$\widehat{V}_C(\boldsymbol{x}) = \underbrace{C_1 \circ_{2,1} w^1(x_1)}_{\in \mathbb{R}^{r_0,r_1}} \cdots \underbrace{C_d \circ_{2,1} w^d(x_d)}_{\in \mathbb{R}^{r_{d-1},r_d}}. \tag{38}$$

Now, in order to utilize stable evaluations, we consider a core representation as discussed in Appendix A.2. To that end, we simplify the discussion and assume that the core position at evaluation is $k = 0$. Consequently, we have

$$C = \mathfrak{C} U_2 \cdots U_d = \mathfrak{C} R_1. \tag{39}$$

Then, by noting that $r_0 = r_d = 1$, the evaluation is realized as

$$\widehat{V}_C(\boldsymbol{x}) = \underbrace{\mathfrak{C} \circ_{2,1} w^1(x_1)}_{\in \mathbb{R}^{1,r_1}} \underbrace{U_2 \circ_{2,1} w^1(x_2)}_{\in \mathbb{R}^{r_1,r_2}} \cdots \underbrace{C_d \circ_{2,1} w^d(x_d)}_{\in \mathbb{R}^{r_{d-1},1}}, \tag{40}$$

which, contracted from right to left, reduces to iterates of matrix-vector multiplications after each $\circ_{2,1}$ contraction is performed first.

### A.4.2 GRADIENT EVALUATION

We define the derivative vector $\dot{w}^i = \dot{w}^i(x_i)$ as

$$[\dot{w}^i(x_i)]_j = (\partial_{x_i}\phi_j^i(x_i))_{j=1}^{m_i} \in \mathbb{R}^{m_i}, \quad i = 1, \ldots, d. \tag{41}$$

Then it holds

$$\partial_{x_i}\widehat{V}_C(\boldsymbol{x}) = C_1 \circ_{2,1} w^1(x_1) \cdots C_i \circ_{2,1} \dot{w}^i(x_i) \cdots C_d \circ_{2,1} w^d(x_d) \tag{42a}$$

$$= \mathfrak{C} \circ_{2,1} w^1(x_1) \cdots U_i \circ_{2,1} \dot{w}^i(x_i) \cdots C_d \circ_{2,1} w^d(x_d). \tag{42b}$$

A naive approach for evaluating gradients would require $d$ separate matrix-vector contractions. However, this procedure can be accelerated by avoiding redundant computations through the use of intermediate stacks. First, motivated by (29), the functional right vector stacks are defined for $k = d, d-1, \ldots, 1$ as

$$R_k(x_{k+1}, \ldots, x_d) = \boldsymbol{U}_{k+1} \circ_{2,1} w^{k+1}(x_{k+1}) \cdots \boldsymbol{U}_d \circ_{2,1} w^d(x_d), \qquad R_k = 1 \text{ for } k = d, \tag{43}$$

where the evaluation is performed from left to right based on matrix-vector multiplications. It holds that $R_k(x_{k+1}, \ldots, x_d) \in \mathbb{R}^{r_k,1}$ is a vector. Note that the recursive relation

$$R_k(x_{k+1}, \ldots, x_d) = \boldsymbol{U}_{k+1} \circ_{2,1} w^{k+1}(x_{k+1}) R_{k+1}(x_{k+2}, \ldots, x_d), \tag{44}$$

holds, avoiding recomputation when building each $R_k$. Moreover, for each $\ell = 2, \ldots, d-1$ we define the intermediate matrix stacks $M_\ell$ as

$$M_2(x_2) = \boldsymbol{U}_2 \circ_{2,1} w^2(x_2), \quad M_\ell(x_2, \ldots, x_\ell) = M_{\ell-1}(x_2, \ldots, x_{\ell-1}) \boldsymbol{U}_\ell \circ_{2,1} w^\ell(x_\ell), \qquad \ell = 3, \ldots, d-1. \tag{45}$$

Again, the recursive relation avoids redundant computations. Consequently, it holds that

$$\partial_{x_1} \widehat{V}_{\boldsymbol{C}}(\boldsymbol{x}) = \mathfrak{C} \circ_{2,1} \dot{w}^1(x_1) R_1(x_2, \ldots, x_d), \tag{46}$$

$$\partial_{x_2} \widehat{V}_{\boldsymbol{C}}(\boldsymbol{x}) = \mathfrak{C} \circ_{2,1} w^1(x_1) \boldsymbol{U}_2 \circ_{2,1} \dot{w}^2(x_i) R_2(x_3, \ldots, x_d). \tag{47}$$

$$\partial_{x_i} \widehat{V}_{\boldsymbol{C}}(\boldsymbol{x}) = \mathfrak{C} \circ_{2,1} w^1(x_1) M_{i-1}(x_2, \ldots, x_{i-1}) \boldsymbol{U}_i \circ_{2,1} \dot{w}^i(x_i) R_i(x_{i+1}, \ldots, x_d), \quad i = 3, \ldots, d. \tag{48}$$

For numerical stability, we first contract

$$M_{i-1}(x_2, \ldots, x_{i-1}) \, \boldsymbol{U}_i \circ_{2,1} \dot{w}^i(x_i) \, R_i(x_{i+1}, \ldots, x_d) \tag{49}$$

in (48), referred to as the orthogonal block. Subsequently, a final contraction is performed with the non-orthogonal block involving the core $\mathfrak{C}$.

This formulation improves the efficiency of gradient evaluation by up to a factor of $d$ compared to naive gradient computations.

### A.4.3 HESSIAN EVALUATION

We define the second derivative vector $\ddot{w}^i = \ddot{w}^i(x_i)$ as

$$[\ddot{w}^i(x_i)]_j = (\partial_{x_i x_i}^2 \phi_j^i(x_i))_{j=1}^{m_i} \in \mathbb{R}^{m_i}, \quad i = 1, \ldots, d. \tag{50}$$

Then it holds

$$\partial_{x_i x_i}^2 \widehat{V}_{\boldsymbol{C}}(\boldsymbol{x}) = \boldsymbol{C}_1 \circ_{2,1} w^1(x_1) \cdots \boldsymbol{C}_i \circ_{2,1} \ddot{w}^i(x_i) \cdots \boldsymbol{C}_d \circ_{2,1} w^d(x_d) \tag{51a}$$

$$= \mathfrak{C} \circ_{2,1} w^1(x_1) \cdots \boldsymbol{U}_i \circ_{2,1} \ddot{w}^i(x_i) \cdots \boldsymbol{C}_d \circ_{2,1} w^d(x_d) \tag{51b}$$

and for $i < j$

$$\partial_{x_i x_j}^2 \widehat{V}_{\boldsymbol{C}}(\boldsymbol{x}) = \boldsymbol{C}_1 \circ_{2,1} w^1(x_1) \cdots \boldsymbol{C}_i \circ_{2,1} \dot{w}^i(x_i) \cdots \boldsymbol{C}_j \circ_{2,1} \dot{w}^j(x_j) \boldsymbol{C}_d \circ_{2,1} w^d(x_d) \tag{52a}$$

$$= \mathfrak{C} \circ_{2,1} w^1(x_1) \cdots \boldsymbol{U}_i \circ_{2,1} \dot{w}^i(x_i) \cdots \boldsymbol{U}_j \circ_{2,1} \dot{w}^j(x_j) \cdots \boldsymbol{C}_d \circ_{2,1} w^d(x_d). \tag{52b}$$

A naive approach for hessian evaluations would require $\mathcal{O}(d^2)$ matrix-vector contractions. The evaluation of diagonal hessian terms can be accelerated in a similar manner to the gradient computation by exploiting the right vector stacks $R_k$ and the mid stack $M_{k-1}$. Moreover, many sub-contractions reappear for different pairs $i < j$ in the remaining terms. Using the right vector stacks $R_k$, we can iteratively construct auxiliary stacks $\dot{R}_k^j$ of the form

$$\dot{R}_i^j(x_{i+1}, \ldots, x_d) = U_{i+1} \circ_{2,1} w^{i+1}(x_{i+1}) \cdots U_j \circ_{2,1} \dot{w}^j(x_j) R_j(x_{j+1}, \ldots, x_d), \tag{53}$$

which can be computed recursively for all $i < j$, while avoiding repeated evaluations of intermediate products of the form

$$U_{i+1} \circ_{2,1} w^{i+1}(x_{i+1}) \cdots U_j \circ_{2,1} \dot{w}^j(x_j). \tag{54}$$

Then the final computation can be carried out as

$$\partial^2_{x_1 x_j} \widehat{V}_{\boldsymbol{C}}(\boldsymbol{x}) = \mathfrak{C} \circ_{2,1} \dot{w}^1(x_1)\dot{R}_1^j(x_2,\dots,x_d), \quad j = 2,\dots,d \tag{55a}$$

$$\partial^2_{x_i x_j} \widehat{V}_{\boldsymbol{C}}(\boldsymbol{x}) = \mathfrak{C} \circ_{2,1} w^1(x_1)M_{i-1}(x_2,\dots,x_{i-1})\boldsymbol{U}_i \circ_{2,1} \dot{w}^i(x_i)\dot{R}_i^j(x_{i+1},\dots,x_d), \quad 1 < i < j. \tag{55b}$$

Again, the contraction in (55b) is performed from left to right, such that the non-orthonormal portion is evaluated in the final step.

This formulation improves the efficiency of Hessian evaluation by up to a factor of $\mathcal{O}(d^2)$ compared to naive Hessian computations.

### A.4.4  MOVING APPROXIMATION DOMAIN AND APPROXIMATE POLICY EXTENSIONS

At timestep $t = t_n$, the empirical loss stated in (10) depends on samples $(\widehat{X}_n^{u,(k)})_{k=1}^K$. Motivated by the idealistic situation that, up to approximation error,

$$\widehat{X}_n^{u,(k)} \sim e^{-V(\cdot,t_n)}, \tag{56}$$

we choose an approximation domain $D_n$ time-step dependent, based on the location of samples. To this end, we define

$$\widehat{a}_{i,n} := \min_{k=1,\dots,K}(\widehat{X}_n^{u,(k)})_i, \qquad \widehat{b}_{i,n} := \max_{k=1,\dots,K}(\widehat{X}_n^{u,(k)})_i, \qquad i = 1,\dots,d. \tag{57}$$

For a domain extension factor $0 \le q < 1$, we then define

$$a_{i,n} := \widehat{a}_{i,n} - q|\widehat{b}_{i,n} - \widehat{a}_{i,n}|, \qquad b_{i,n} := \widehat{b}_{i,n} + q|\widehat{b}_{i,n} - \widehat{a}_{i,n}|, \qquad i = 1,\dots,d. \tag{58}$$

In our experiments, we use $q = 0.1$. Finally, we introduce the adapted tensor domain at time step $t = t_n$ as

$$D_n := \underset{i=1}{\overset{d}{\times}} [a_{i,n}, b_{i,n}]. \tag{59}$$

Consequently, the tensor basis set $\mathcal{B}_m$ is time-step dependent, i.e. $\mathcal{B}_m = \mathcal{B}_m(D_n)$.

For the reverse sampling dynamics, trajectories may leave the prescribed evaluation domain $D_n$ at time $t = t_n$, rendering the evaluation of the gradient entering the policy ill-defined, since the value function is approximated using basis functions from $\mathcal{B}_{\boldsymbol{m}}(D_n)$. To address this issue, we consider a simple affine linear extension strategy. Let $0 \le p < 1$ denote a domain shrinkage factor and define the shrunken tensor domain

$$D_{n,p} = \underset{i=1}{\overset{d}{\times}} [a_{i,n,p}, b_{i,n,p}], \tag{60}$$

with

$$a_{i,n,p} = a_{i,n} + p|b_{i,n} - a_{i,n}|, \quad b_{i,n,p} = b_{i,n} - p|b_{i,n} - a_{i,n}|. \tag{61}$$

In our numerical experiment, we used $p = 0.1$ in accordance to $q$ above.

Let $\boldsymbol{x} \in \mathbb{R}^d$. We define the *extended gradient* evaluation of a function $v \in \operatorname{span}\mathcal{B}_{\boldsymbol{m}}(D_n)$ as follows. Let $\Pi_{D_{n,p}}\boldsymbol{x}$ denote the projection of $\boldsymbol{x}$ onto $D_{n,p}$. Then

$$\nabla_{\mathrm{ext}}v(\boldsymbol{x},t_n) := \nabla v(\Pi_{D_{n,p}}\boldsymbol{x}) + H_v(\boldsymbol{x} - \Pi_{D_{n,p}}\boldsymbol{x}), \tag{62}$$

with the well defined gradient $\nabla v$ and Hessian $H_v$ for points within $D_n$. In practice, the computation of the Hessian will only be realized if $\boldsymbol{x} \ne \Pi_{D_{n,p}}\boldsymbol{x}$.

*Remark* A.1. We note that during the sampling stage, the vast majority of trajectories remain within the predefined moving domains $D_n$. As a result, only a small fraction of trajectories require evaluation of the Hessian, thereby limiting potential computational overhead. Moreover, as discussed above, the application of Hessians in the xTT format can in any case be carried out with high efficiency.

*Remark* A.2. The choice of $p > 0$ together with the design of an affine-type extension is motivated as follows. First, the affine structure ensures that trajectories leaving the computational domain are mapped back in a controlled manner while still being guided in an approximately correct direction. Second, potential boundary-induced oscillations are absorbed in the exterior region parameterized by $q > 0$ and are subsequently damped by the shrinkage parameter $p > 0$.

An illustration of our extension mechanism is provided in Figure 8.

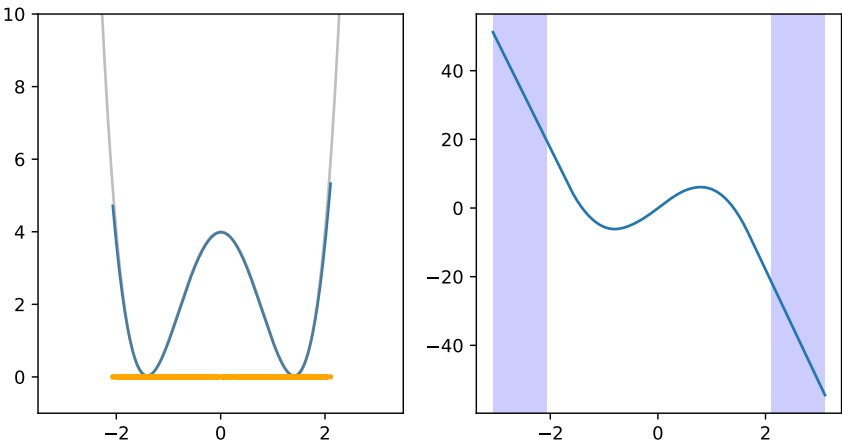

*Figure 8.* Illustration of the linear extension procedure. Left: the log target density (gray), the approximated value function at time step $t = t_{N-1}$ (blue), and the training samples used for the approximation (orange). Right: the resulting approximate policy, where the shaded region indicates the domain in which the linear extension is activated.

### A.4.5    REPRESENTATION CHANGE ON DIFFERENT DOMAINS

In view of the iterative structure in Algorithm 1, multiple instances of minimization problems associated with the explicit BSDE loss in (9) must be solved. Each optimization problem requires an initial guess. Motivated by the continuous evolution of the value function $V$ in time, a natural choice for the initial guess at time step $t = t_n$ is the approximation of $V$ obtained at the previous time step $t = t_{n+1}$. However, since we employ a sequence of adaptively moving domains $(D_n)_n$, in general it holds that $D_{n+1} \neq D_n$.

Consequently, a direct copy-paste of the extended tensor train representation is not valid. In fact, the basis functions at time step $t = t_n$ change, and accordingly the coefficient tensor must be updated.

Fortunately, the extended tensor train format admits an efficient change of representation between different domains, which we introduce next.

Let $D^1 = \times_{i=1}^{d} [a_{i,1}, b_{i,1}]$ and $D^2 = \times_{i=1}^{d} [a_{i,2}, b_{i,2}]$ be two tensor domains with non-trivial overlap $O = D^1 \cap D^2$. Then

$$O = \underset{i=1}{\overset{d}{\times}} [\max\{a_{i,1}, a_{i,2}\}, \min\{b_{i,1}, b_{i,2}\}] := \underset{i=1}{\overset{d}{\times}} [a_i, b_i] \tag{63}$$

is a tensor domain as well. Moreover, for $\boldsymbol{m}^1 = (m_1^1, \dots, m_d^1), \boldsymbol{m}^2 = (m_1^2, \dots, m_d^2) \in \mathbb{N}^d$ consider two tensor basis sets with respect to $D^1$ and $D^2$:

$$\mathcal{B}_{\boldsymbol{m}^\ell}^\ell := \left\{ \phi_{\boldsymbol{\alpha}}^\ell := \bigotimes_{i=1}^{d} \phi_{\alpha_i}^{i,\ell} \;\middle|\; \boldsymbol{\alpha} \in [\boldsymbol{m}^\ell] \right\}, \quad \ell = 1, 2. \tag{64}$$

We note that we omit the orthonormalization conditions at this point, as they are irrelevant for the subsequent discussion.

We then adopt a Galerkin-type projection ansatz for the change of representation from basis functions defined on $D^1$ to basis functions defined on $D^2$. To this end, consider the tensor-product Hilbert space

$$\mathcal{H}(O) = \bigotimes_{i=1}^{d} \mathcal{H}_i(a_i, b_i), \tag{65}$$

with univariate inner products $(\cdot, \cdot)_{\mathcal{H}_i(a_i, b_i)}$ for $i = 1, \dots, d$.

Let $i = 1, \dots, d$ be fixed. We define the *overlap Gramian matrix* for basis functions from $\mathcal{B}_{\boldsymbol{m}^2}^2$ as

$$[G_i^2]_{jk} = (\phi_j^{i,2}, \phi_k^{i,2})_{\mathcal{H}_i(a_i, b_i)}, \qquad j, k = 1, \dots, m_i^2, \tag{66}$$

and the *overlapping interaction matrix*

$$[M_i]_{jk} = (\phi_j^{i,2}, \phi_k^{i,1})_{\mathcal{H}_i(a_i,b_i)}, \qquad j,k = 1,\ldots,m_i^2. \tag{67}$$

Then, assuming that $G_i^2$ is invertible, we define the univariate representation change operator

$$T_i := [G_i^2]^{-1} M_i \in \mathbb{R}^{m_i^2, m_i^1}. \tag{68}$$

*Application*: Now, let $\widehat{V}_{C^1}^1(\boldsymbol{x}) = \boldsymbol{C}^1[\Phi^1(\boldsymbol{x})]$ be an xTT on $D^1$ with basis functions from $\mathcal{B}_{\boldsymbol{m}^1}^1$ and a tensor train given as

$$\boldsymbol{C}^1 = \boldsymbol{C}_1^1 \cdots \boldsymbol{C}_d^1. \tag{69}$$

Then, we define a new xTT

$$\widehat{V}_{C^2}^2 = \boldsymbol{C}^2[\Phi^2(\boldsymbol{x})] \tag{70}$$

with

$$\boldsymbol{C}^2 = \boldsymbol{C}_1^2 \cdots \boldsymbol{C}_d^2, \quad \boldsymbol{C}_i^2 := T_i \circ_{2,2} \boldsymbol{C}_i^1, \tag{71}$$

That is, $T_i$ acts on the second mode of $\boldsymbol{C}_i^1$, which corresponds to the basis discretization.

In practice, we choose $\mathcal{H}_i(O)$ in accordance with the Hilbert space structures $\mathcal{H}_i(a_{i,1}, b_{i,1})$ and $\mathcal{H}_i(a_{i,2}, b_{i,2})$. For instance, if we set $\mathcal{H}_i(a_{i,1}, b_{i,1}) = H^2(a_{i,1}, b_{i,1})$, then we choose the corresponding univariate Hilbert spaces on the overlapping intervals $[a_i, b_i]$ to be Sobolev spaces of order 2, i.e. $H^2(a_i, b_i)$. In this case, the resulting tensor-product space becomes $\mathcal{H}(O) = H_{\mathrm{mix}}^2(O)$. Furthermore, the scalar products in (66) and (67) can be computed via efficient one-dimensional numerical integration at negligible computational cost.

## A.5 Empirical loss minimization via ALS

We are now ready to formulate the alternating least squares (ALS) scheme, originally introduced in Holtz et al. (2012a). As a first, more accessible objective, we consider the classical empirical $L^2$ loss and subsequently extend the discussion to our BSDE loss.

### A.5.1 ALS for empirical $L^2$ losses

Let $D$ be a fixed tensor domain and assume we have samples $X^{(k)} \in D$ and values $y^{(k)}$. Furthermore, denote by $X_i^{(k)}$ the $i$-th component of $X^{(k)}$. Our goal is to fit a extended tensor train $v_C \in \mathcal{X}_{\boldsymbol{R}}^{\boldsymbol{m}}$ to the data using a regularized empirical loss of the form

$$\mathcal{L}(v) := \frac{1}{K} \sum_{k=1}^{K} |v(X^{(k)}) - y^{(k)}|^2 + \tau \|v\|_{\mathcal{H}}^2. \tag{72}$$

In the spirit of Appendix A.4.1, we define the univariate feature matrices $W_i \in \mathbb{R}^{K,m_i}$ with

$$[W_i]_{k\ell} = \phi_\ell^i((X^{(k)})_i), \qquad k = 1,\ldots,K, \quad \ell = 1,\ldots,m_i. \tag{73}$$

Now, assume that the tensor train $\boldsymbol{C} = \boldsymbol{U}_1 \cdots \boldsymbol{U}_{i-1} \mathfrak{C} \boldsymbol{U}_{i+1} \cdots \boldsymbol{U}_d$ has its core at position $i$. Using subsequent contractions $\boldsymbol{U}_i \circ_{2,2} W_i \in \mathbb{R}^{r_{i-1},K,r_i}$ and defining $Y = (y^{(k)})_k \in \mathbb{R}^{K,1}$, we can write

$$\mathcal{L}(v_C) = \frac{1}{K} \sum_{k=1}^{K} |\boldsymbol{C}[\Phi(X^{(k)})] - y^{(k)}|^2 + \tau \|\boldsymbol{C}\|_{\mathrm{F}}^2 \tag{74a}$$

$$= \frac{1}{K} \|\boldsymbol{U}_1 \circ_{2,2} W_1 \cdots \boldsymbol{U}_{i-1} \circ_{2,2} W_{i-1} \mathfrak{C} \circ_{2,2} W_i \cdots \boldsymbol{U}_d \circ_{2,2} W_d - Y\|_2^2 + \tau \|\mathfrak{C}\|_{\mathrm{F}}^2 \tag{74b}$$

$$= \frac{1}{K} \|\boldsymbol{L}_i \mathfrak{C} \circ_{2,2} W_i \boldsymbol{R}_i - Y\|_2^2 + \tau \|\mathfrak{C}\|_{\mathrm{F}}^2, \tag{74c}$$

with order three tensors (in fact matrices) given as

$$\boldsymbol{L}_i := \boldsymbol{U}_1 \circ_{2,2} W_1 \cdots \boldsymbol{U}_{i-1} \circ_{2,2} W_{i-1} \in \mathbb{R}^{1,K,r_{i-1}}, \tag{75}$$

$$\boldsymbol{R}_i := \boldsymbol{U}_{i+1} \circ_{2,2} W_{i+1} \cdots \boldsymbol{U}_d \circ_{2,2} W_d \in \mathbb{R}^{r_i,K,1}. \tag{76}$$

Now, the key observation is that $\boldsymbol{L}_i$, $\boldsymbol{R}_i$, and $W_i$ define a linear operator acting on $\mathfrak{C}$. To make this precise, let $L_i \in \mathbb{R}^{K, r_{i-1}}$ and $R_i \in \mathbb{R}^{r_i, K}$ denote the matrix representations of the squeezed tensors $\boldsymbol{L}_i$ and $\boldsymbol{R}_i$, respectively, and let $y \in \mathbb{R}^K$ denote the squeezed version of the matrix $Y$. Then, we can define the matrix

$$A_i := R_i^\top \otimes W_i \otimes L_i \in \mathbb{R}^{K, r_{i-1} m r_i}. \tag{77}$$

Let $c := \mathrm{vec}(\mathfrak{C})$, then it holds for $y = (y^{(k)})_k \in \mathbb{R}^K$

$$\mathcal{L}(v_{\boldsymbol{C}}) = \frac{1}{K} \|A_i c - y\|_2^2 + \tau \|c\|_2^2. \tag{78}$$

Now, in a sweeping manner, moving forward and backward over each TT component, we shift the core to position $i$, solve the resulting least-squares problem (78), and update the core by reshaping the corresponding solution. The basic principle is summarized in Algorithm 2.

---

**Algorithm 2** ALS for tensor train minimization with orthogonal structure

---

1: **Input:** Initial components $\{\boldsymbol{C}_1^{(0)}, \ldots, \boldsymbol{C}_d^{(0)}\}$, regularization magnitude $\tau > 0$, maximal iterations $I$, error stop criteria $\varepsilon$, data $(X^{(k)})_k, y \in \mathbb{R}^K$.
2: **Output:** Optimized TT representation $\boldsymbol{C} = \boldsymbol{C}_1 \cdots \boldsymbol{C}_d$ for xTT $v_{\boldsymbol{C}} \in \mathcal{X}_{\boldsymbol{r}}^{\boldsymbol{m}}$
3: Evaluate feature matrices $W_i$ from (73).
4: Set $n = 0$
5: **repeat**
6:     Set $n = n + 1$
7:     **for** $i = 1, 2, \ldots, d, d-1, \ldots, 2$ **do**
8:         Set TT core to position $i$ yielding $\boldsymbol{C} = \boldsymbol{U}_1^{(n)} \cdots \boldsymbol{U}_{i-1}^{(n)} \mathfrak{C} \boldsymbol{U}_{i+1}^{(n)} \cdots \boldsymbol{U}_d^{(n)}$
9:         Construct $A_i^{(n)}$ from left and right orthogonal components $\boldsymbol{L}_i^{(n)}, \boldsymbol{R}_i^{(n)}$ and $W_i$ as in (77).
10:         Solve micro-step problem:
11:         $c^* = \arg\min_{c \in \mathbb{R}^{r_{i-1} m_i r_i}} \|A_i^{(n)} c - y\|_2^2 + \tau \|c\|_F^2$
12:         Reshape $c^*$ to $\mathfrak{C}^* \in \mathbb{R}^{r_{i-1}, m_i, r_i}$ and update core at position $i$ to $\mathfrak{C} = \mathfrak{C}^*$.
13:     **end for**
14: **until** error stop criteria is met or $n \geq N$

---

For the interested reader, we note that the subsequent shift of the core position within the for-loop of Algorithm 2 (the so-called *forward-backward sweep*) requires only a single local singular value decomposition, as described in Appendix A.2. In particular, all components in the current representation remain unchanged except for two factors: the former core and the newly shifted core. This observation enables significant speedups of the algorithm, since the left component $\boldsymbol{L}_i$ and the right component $\boldsymbol{R}_i$ can be constructed during the backward sweep ($i = d, d-1, \ldots, 2$) and forward sweep ($i = 1, \ldots, d-1$) using the stack-based principle discussed in Appendix A.4. This avoids redundant contractions within a single sweep.

As a stopping criterion, we choose $\varepsilon > 0$ such that the relative loss satisfies

$$\frac{1}{K} \frac{\sum_{k=1}^K \left| v(X^{(k)}) - y^{(k)} \right|^2}{\|y\|_2^2} \leq \varepsilon. \tag{79}$$

This quantity can be efficiently evaluated at each outer iteration $n$ and micro step $i$ as

$$\frac{1}{K} \sum_{k=1}^K \left| v(X^{(k)}) - y^{(k)} \right|^2 = \frac{1}{K} \left\| A_i^{(n)} \mathrm{vec}(\mathfrak{C}) - y \right\|_2^2. \tag{80}$$

### A.5.2 ALS FOR BSDE LOSSES WITH GENERAL BASIS FUNCTIONS

We now extend the ALS approach to the BSDE loss function in the general setting where derivatives of the basis functions are not necessarily contained in the span of the original basis. This situation arises, for instance, for $\mathcal{C}^s$ spline spaces of polynomial degree $p \geq s + 2$.

We begin with the BSDE loss

$$\widehat{\mathcal{L}}_n^K(\widehat{V}_n) = \frac{1}{K} \sum_{k=1}^{K} \left( (\text{Id} + \Sigma_n^{(k)} \cdot \nabla) \widehat{V}_n(\widehat{X}_n^{u,(k)}) - y_{n+1}^{(k)} \right)^2. \tag{81}$$

Let $\widehat{V}_n$ be represented by an xTT $v_{\boldsymbol{C}} \in \mathcal{X}_{\boldsymbol{r}}^{\boldsymbol{m}}$ with tensor train with core position at $j$ given as

$$\boldsymbol{C} = \boldsymbol{U}_1 \cdots \boldsymbol{U}_{j-1} \mathfrak{C} \boldsymbol{U}_{j+1} \cdots \boldsymbol{U}_d. \tag{82}$$

For each sample $k = 1, \ldots, K$, define the basis evaluations as

$$w_j^{(k)}[\alpha_j] := \phi_{\alpha_j}^j(X_{n,j}^{u,(k)}), \tag{83}$$

$$\dot{w}_j^{(k)}[\alpha_j] := \partial_x \phi_{\alpha_j}^j(X_{n,j}^{u,(k)}), \quad \forall j = 1, \ldots, d, \tag{84}$$

where we omit the dependency on $u$ and $n$ for notational convenience. We introduce the feature matrices $W_i, \dot{W}_i \in \mathbb{R}^{K,m_i}$ as

$$[W_i]_{k\ell} = w_i^{(k)}[\ell], \qquad [\dot{W}_i]_{k\ell} = \partial_{x_i} \dot{w}_i^{(k)}[\ell]. \tag{85}$$

Further, we introduce the weighted feature matrices for $i = 0, \ldots, d$ and $j = 1, \ldots, d$ as

$$W_{ij} = \begin{cases} W_j & \text{if } i \neq j, \\ \Sigma_j^n * \dot{W}_j & \text{if } i = j, \end{cases} \tag{86}$$

where the operator $*$ refers to pointwise multiplication in the sample direction slice for each $k = 1, \ldots, K$ with the vector $\Sigma_i^n \in \mathbb{R}^K$ with

$$(\Sigma_i^n)_k = [\Sigma^{n,(k)}]_i. \tag{87}$$

This allows us to define rank 1 tensors:

$$\boldsymbol{W}_0 = W_{0,1} \otimes \cdots \otimes W_{0,d}, \tag{88}$$

$$\dot{\boldsymbol{W}}_i = W_{i,1} \otimes \cdots \otimes W_{i,d}. \tag{89}$$

Now, with the core at position $j$, we define the left and right contractions for each $i = 0, \ldots, d$ as

$$\boldsymbol{L}_{i,j} = \boldsymbol{U}_1 \circ_{2,2} W_{i1} \cdots \boldsymbol{U}_{j-1} \circ_{2,2} W_{i,j-1} \in \mathbb{R}^{1,K,r_{i-1}}, \tag{90}$$

$$\boldsymbol{R}_{i,j} = \boldsymbol{U}_{j+1} \circ_{2,2} W_{i,j+1} \cdots \boldsymbol{U}_d \circ_{2,2} W_{i,d} \in \mathbb{R}^{r_i,K,1}, \tag{91}$$

with the conventions $L_{i,1} = \mathbf{1}$ and $R_{i,d} = \mathbf{1}$.

The BSDE operator applied to the tensor train for $(X_{n,j}^{u,(k)}) \in \mathbb{R}^{K,d}$ becomes

$$(\text{Id} + \Sigma^{n,(k)} \cdot \nabla) \widehat{V}^n((X_{n,j}^{u,(k)})) = \boldsymbol{L}_{0,j}(\mathfrak{C} \circ_{2,2} W_{0j}) \boldsymbol{R}_{0,j} + \sum_{i=1}^{d} \boldsymbol{L}_{i,j}(\mathfrak{C} \circ_{2,2} W_{ij}) \boldsymbol{R}_{i,j} \tag{92a}$$

$$= \sum_{i=0}^{d} \boldsymbol{L}_{i,j}(\mathfrak{C} \circ_{2,2} W_{ij}) \boldsymbol{R}_{i,j}. \tag{92b}$$

Analogously to the derivation in (74c) for the vector $y = (y_{n+1}^{(k)})_k \in \mathbb{R}^{K,1}$ we obtain

$$\widehat{\mathcal{L}}_n^K(v_{\boldsymbol{C}}) = \frac{1}{K} \left\| \sum_{i=0}^{d} \boldsymbol{L}_{i,j}(\mathfrak{C} \circ_{2,2} W_{ij}) \boldsymbol{R}_{i,j} - y \right\|_2^2 + \tau \|\mathfrak{C}\|_{\text{F}}^2. \tag{93}$$

Recall at this point that $\boldsymbol{L}_{i,j}, W_{i,j}$ and $\boldsymbol{R}_{i,j}$ depend on $u$ and $n$. We can define the matrix

$$A_j := \sum_{i=0}^{d} R_{i,j}^{\top} \otimes W_{i,j} \otimes L_{i,j} \in \mathbb{R}^{K, r_{i-1} m r_i}, \tag{94}$$

where $R_{i,j}$ and $L_{i,j}$ again denote the matrices obtained from squeezing the last mode dimension of $\boldsymbol{R}_{i,j}$ and the first mode dimension of $\boldsymbol{L}_{i,j}$.

Then, in the spirit of (77), it holds that

$$\widehat{\mathcal{L}}_n^K(v_{\boldsymbol{C}}) = \frac{1}{K} \|A_j \operatorname{vec}(\mathfrak{C}) - y\|_2^2 + \tau \| \operatorname{vec}(\mathfrak{C})\|_2^2. \tag{95}$$

---

**Algorithm 3** ALS for BSDE with general basis functions

---

1: **Input:** Initial TT cores $\{\boldsymbol{U}_1^{(0)}, \ldots, \boldsymbol{U}_d^{(0)}\}$, regularization $\tau > 0$, maximal iterations $I$, tolerance $\varepsilon$, samples $(\widehat{X}_n^{u,(k)}, \Sigma_n^{(k)}, y_{n+1}^{(k)})_k$
2: **Output:** Optimized TT representation for $\widehat{V}_n$
3: Precompute feature evaluation matrices $W_j, \dot{W}_j$ from (85) and define $W_{ij}$ from (86).
4: Define the vector $y = (y_{n+1}^{(k)})_k$.
5: Set $\ell = 0$.
6: **repeat**
7:     **for** $j = 1, 2, \ldots, d, d-1, \ldots, 2$ **do**
8:         Set TT core to position $j$: $\boldsymbol{C} = \boldsymbol{U}_1^{(\ell)} \cdots \boldsymbol{U}_{j-1}^{(\ell)} \mathfrak{C} \boldsymbol{U}_{j+1}^{(\ell)} \cdots \boldsymbol{U}_d^{(\ell)}$.
9:         Compute $\boldsymbol{L}_{i,j}, \boldsymbol{R}_{i,j}$ using a stack principle for each $i = 0, \ldots, d$.
10:        Build micro step system matrix $A_j$ from (94).
11:       Solve: $c^* = \arg\min_c \|A_j c - y\|_2^2 + \tau \|c\|_F^2$.
12:       Reshape $c^*$ to update core $\boldsymbol{U}_j^{(\ell)} = \mathfrak{C}$.
13:     **end for**
14:     Set $\ell = \ell + 1$, all TT components have been updated.
15:     Compute loss: $\mathcal{L}^{(\ell)} = \frac{1}{K} \|A_j c - b\|_2^2$.
16: **until** $|\mathcal{L}^{(\ell)} - \mathcal{L}^{(\ell-1)}| < \varepsilon$ or $\ell \geq I$.

---

The computational efficiency is maintained through the stack principle: during the forward sweep ($j = 1, \ldots, d$), the left contractions $L_{i,j}^{(k)}$ can be updated incrementally, and during the backward sweep ($j = d, \ldots, 2$), the right contractions $R_{i,j}^{(k)}$ can be updated incrementally.

## A.6 Adaptivity

Our algorithm features adaptivity with respect to the regularization parameter $\tau_n$ (see Appendix A.6.1), the underlying xTT ranks (see Appendix A.6.2), and the basis degrees (see Appendix A.6.3) used in the functional tensor train approximation.

### A.6.1 Choice of regularization magnitude $\tau_n$

Recall that our regularized loss in (15) is given as

$$\widehat{\mathcal{L}}_n^K(\widehat{V}_{\boldsymbol{C}}) + \tau_n \|\widehat{V}_{\boldsymbol{C}}\|_{\mathcal{H}}^2 = \frac{1}{K} \sum_{k=1}^{K} \left( \boldsymbol{C}[\Phi(\widehat{X}_n^{u,(k)})] + \Sigma_n^{(k)} \cdot \nabla \boldsymbol{C}[\Phi(\widehat{X}_n^{u,(k)})] - y_{n+1}^{(k)} \right)^2 + \tau_n \|\boldsymbol{C}\|_{\mathrm{F}}^2. \tag{96}$$

If the tensor train $\boldsymbol{C}$ is represented as

$$\boldsymbol{C} = \boldsymbol{C}_1 \boldsymbol{C}_2 \cdots \boldsymbol{C}_d = \boldsymbol{U}_1 \cdots \boldsymbol{U}_{i-1} \mathfrak{C} \boldsymbol{U}_{i+1} \cdots \boldsymbol{U}_d \tag{97}$$

with orthogonal order three tensors $U_j \in \mathbb{R}^{r_{j-1}, m_j, r_j}$ for $j = 1, \ldots, i-1, i+1, \ldots d$ and a *core* $\mathfrak{C} \in \mathbb{R}^{r_{i-1}, m_i, r_i}$ at position $i$, then, we have that

$$\widehat{\mathcal{L}}_n^K(\widehat{V}_{\boldsymbol{C}}) + \tau_n \|\widehat{V}_{\boldsymbol{C}}\|_{\mathcal{H}}^2 = \|A_{i,n}^u \operatorname{vec}(\mathfrak{C}) - y_{n+1}\|_2^2 + \tau_n \|\mathfrak{C}\|_F^2, \tag{98}$$

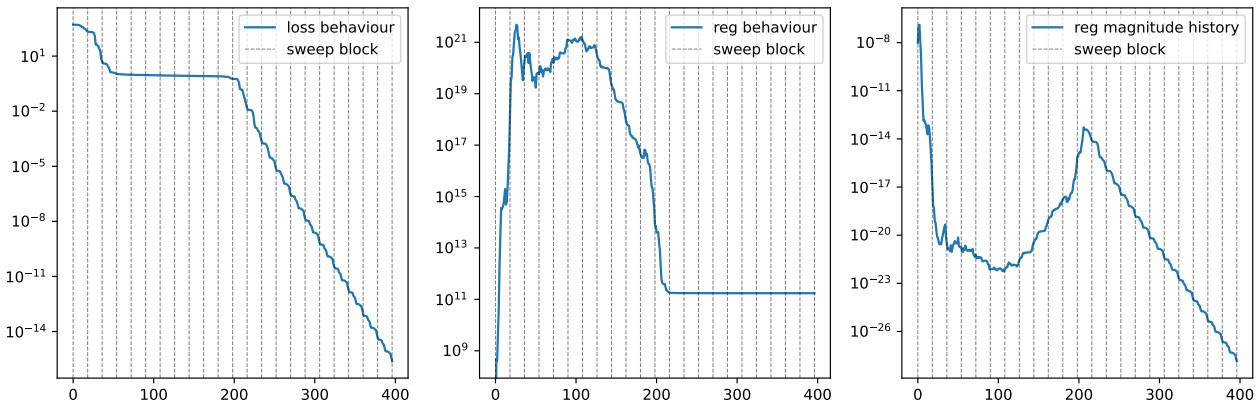

*Figure 9.* Example of adaptively determining the regularization parameter $\tau_n$ for the Multiwell problem in dimension $d = 10$. Left: the empirical loss value $\widehat{\mathcal{L}}_n^K(\widehat{V}_C)$. Middle: the value $\|\widehat{V}_C\|_{\mathcal{H}}^2$. Right: the adaptive evolution of $\tau_n$.

for a matrix $A_{i,n}^u \in \mathbb{R}^{K, r_{i-1} m_i r_i}$. In particular, the original empirical BSDE loss can be efficiently evaluated at each micro step of the proposed ALS algorithm as

$$\widehat{\mathcal{L}}_n^K(\widehat{V}_C) = \left\| A_{i,n}^u \operatorname{vec}(\mathfrak{C}) - y_{n+1} \right\|_2^2. \tag{99}$$

Moreover, the regularization in the $\mathcal{H}$-norm reduces, via the Parseval identity and the tensor train core representation, to a single Frobenius norm evaluation of the core tensor $\mathfrak{C}$.

In our experiments, we observed that a naive choice of $\tau_n$, e.g. $\tau_n \equiv 10^{-8}$ independently of the dimension $d$, does not yield a convergent scheme. This can be understood through the behavior of the Hilbert norm $\|\widehat{V}_C\|_{\mathcal{H}}^2$. Our objective is that, at each time step $t = t_n$, the minimizer $\widehat{V}_C$ approximates the true value function $V(\cdot, t_n)$. However, the quantity

$$t \mapsto \|V(\cdot, t)\|_{\mathcal{H}}^2 \tag{100}$$

may vary significantly over time and dimension $d$, and in our experiments it appears to grow exponentially with $d$. Consequently, the parameter $\tau_n$ must compensate for the scaling of the $\|\cdot\|_{\mathcal{H}}^2$ term.

In our implementation, we fix a relative weighting parameter $\gamma = 0.1$ and determine $\tau_n$ adaptively as follows. At the final time step $n = N$, we initialize $\tau_N$. Now assume that at time $t = t_n$, during the $k$-th sweep and micro step $i$, we obtain an updated tensor train representation $C$, in particular updating the core at position $i$ using the current value of $\tau_n$. This allows us to compute both $\|A_{i,n}^u \operatorname{vec}(\mathfrak{C}) - y_{n+1}\|_2^2$ and $\|\mathfrak{C}\|_F^2$. We then update

$$\tau_n = \gamma \frac{\|A_{i,n}^u \operatorname{vec}(\mathfrak{C}) - y_{n+1}\|_2^2}{\|\mathfrak{C}\|_F^2}, \tag{101}$$

and proceed to the next micro step. This choice ensures that

$$\tau_n \|\mathfrak{C}\|_F^2 = \gamma \|A_{i,n}^u \operatorname{vec}(\mathfrak{C}) - y_{n+1}\|_2^2, \tag{102}$$

i.e. the regularization term is of order $\gamma$ relative to the data fidelity term. In all experiments, we use $\gamma = 0.1$.

Finally, when transitioning from time step $t = t_{n+1}$ to $t = t_n$, the previously obtained value $\tau_{n+1}$ serves as a natural initialization for $\tau_n$, since empirically $\|V(\cdot, t_n)\|_{\mathcal{H}}^2 \approx \|V(\cdot, t_{n+1})\|_{\mathcal{H}}^2$.

It is worth noting that choosing $\gamma = 0$, and hence $\tau_n = 0$, does not yield a robust algorithm, since oscillations in the derivatives still need to be controlled. In our experiments, we found that non-negligible values of $\tau_n$ are necessary to stabilize the learning dynamics; see Figure 9.

*Remark* A.3. An alternative approach would be to construct the tensor basis orthonormally with respect to the (empirical) weighted Hilbert spaces, which corresponds to orthonormalization using empirical Gram matrices. This is expected to provide better control over the magnitude of $\|\cdot\|_{\mathcal{H}}^2$, thereby reducing the sensitivity of the algorithm to the choice of $\tau_n$. We leave this direction for future work.

### A.6.2 Adaptivity for Rank Design

An important feature of our algorithm is the adaptive selection of TT ranks, i.e. determining suitable ranks required to achieve a prescribed approximation quality of the value function. Based on the structure of the HJB PDE (4), we expect that the functional tensor train (FTT) rank of the value function $V(\cdot, t)$ varies over time. Intuitively, the nonlinear term $\|\sigma^\top \nabla V\|^2$ may induce a temporary increase in rank, while the diffusion term acts over time to reduce the FTT rank $\overline{r} \in \mathbb{N}^{d-1}$ toward the TT rank of the log-density of the invariant distribution of the reversing process. For instance, if the reversing process is an Ornstein-Uhlenbeck (OU) process, then the invariant distribution is standard normal and the limiting FTT rank becomes $\mathbf{2} = (2, \ldots, 2) \in \mathbb{N}^{d-1}$, see Gruhlke et al. (2026).

Recall the singular value decomposition of a tensor train component $\boldsymbol{C}_i$ after reshaping it into a matrix $C_i$, as described in Appendix A.2, namely

$$C_i = U_i \Sigma_i V_i^\top. \tag{103}$$

The decay pattern of the singular values in $\Sigma_i$, denoted by $\sigma_1^i, \ldots, \sigma_{r_i}^i$, provides valuable guidance for adapting the rank $r_i$. In particular, if all singular values are of comparable and non-negligible magnitude, this indicates that a larger rank $r_i$ may be necessary. Conversely, if the tail singular values $\sigma_{k+1}^i, \ldots, \sigma_{r_i}^i$ are close to zero, the effective rank can be safely reduced to $r_i = k$. It is furthermore advisable to assess the singular values relative to the scale of the loss function, as this helps to avoid overfitting to sampling noise.

We now apply a high-order singular value decomposition (HOSVD, see Grasedyck (2010); Oseledets (2011)) to $\boldsymbol{C}$. To this end, fix an index $1 \leq k \leq d$, referred to as the *core position*. For each $i = 1, \ldots, k-1$, we proceed iteratively as follows. We reshape the $i$-th TT component $\boldsymbol{C}_i \in \mathbb{R}^{r_{i-1}, m_i, r_i}$ into a matrix

$$C_i \in \mathbb{R}^{r_{i-1}, m_i r_i}, \tag{104}$$

compute its singular value decomposition

$$C_i = U_i \Sigma_i V_i^\top, \tag{105}$$

and absorb the non-orthonormal factor $\Sigma_i V_i^\top$ into the next TT component by redefining

$$\boldsymbol{C}_{i+1} \leftarrow \Sigma_i V_i^\top \, \boldsymbol{C}_{i+1}. \tag{106}$$

Analogously, for $i = d, d-1, \ldots, k+1$, we iteratively compute the singular value decompositions

$$C_i = U_i \Sigma_i V_i^\top, \tag{107}$$

and absorb the factor $U_i \Sigma_i$ into the preceding TT component by contracting it from the right with $C_{i-1}$. Finally, we reshape each matrix $U_i \in \mathbb{R}^{r_{i-1}, m_i r_i}$ into an order-three tensor

$$\boldsymbol{U}_i \in \mathbb{R}^{r_{i-1}, m_i, r_i}, \quad \text{for } i = 1, \ldots, k-1, \tag{108}$$

and each matrix $V_i^\top \in \mathbb{R}^{r_{i-1}, m_i r_i}$ into an order-three tensor

$$\boldsymbol{U}_i \in \mathbb{R}^{r_{i-1}, m_i, r_i}, \quad \text{for } i = d, d-1, \ldots, k+1. \tag{109}$$

Then, the TT representation satisfies

$$\boldsymbol{C} \;=\; \boldsymbol{U}_1 \cdots \boldsymbol{U}_{k-1} \, \boldsymbol{C}_k \, \boldsymbol{U}_{k+1} \cdots \boldsymbol{U}_d. \tag{110}$$

We propose a simple adaptive rank design in Algorithm 4.

Additionally, rank adjustments can be guided by target loss values, taking into account the level of sample noise. In our experiments, we did not employ this approach. Since the algorithm requires multiple optimization instances per time step, we introduce a rank update frequency to regulate the application of Algorithm 4.

In our experiments, we typically set the rank update frequency to 10 or 20 and choose the relative singular value magnitude parameter as $\delta = 10^{-4}$.

---

**Algorithm 4** Adaptive rank adjustment for tensor train learning

---

1: **Input:** Initial TT rank $\boldsymbol{r}^{(0)} = (r_1^{(0)}, \ldots, r_{d-1}^{(0)})$, threshold $\delta > 0$, maximum iterations $N$
2: **Output:** Optimized TT representation with adapted ranks
3: Initialize $\rho_i \leftarrow \emptyset$ for $i = 1, \ldots, d-1$                                              {Set of previously seen ranks}
4: fixed $\leftarrow \emptyset$                                                     {Set of fixed rank indices}
5: **repeat**
6:     Optimize loss with current fixed TT rank $\boldsymbol{r}$                            {Using ALS or other method}
7:     **for** $i = 1$ to $d-1$ **do**
8:         **if** $i \notin$ fixed **then**
9:             Perform HOSVD and get singular values $\sigma_1^i \geq \cdots \geq \sigma_{r_i}^i$
10:             $k_i \leftarrow \min\{\ell \mid \sigma_\ell^i < \delta \sigma_1^i\}$
11:             **if** $k_i = r_i$ **then**
12:                 $r_i^{\text{new}} \leftarrow r_i + 1$                                          {Increase rank}
13:                 Add random orthonormal direction to $U_i$ and $U_{i+1}$
14:             **else if** $k_i < r_i$ **then**
15:                 $r_i^{\text{new}} \leftarrow k_i$                                            {Decrease rank}
16:             **end if**
17:             **if** $r_i^{\text{new}} \in \rho_i$ **then**
18:                 fixed $\leftarrow$ fixed $\cup \{i\}$                                    {Fix this rank}
19:             **else**
20:                 $\rho_i \leftarrow \rho_i \cup \{r_i^{\text{new}}\}$
21:                 $r_i \leftarrow r_i^{\text{new}}$
22:             **end if**
23:         **end if**
24:     **end for**
25: **until** fixed $= \{1, \ldots, d-1\}$ or maximum iterations reached

---

### A.6.3   ADAPTIVITY FOR BASIS DEGREES

The third adaptivity component of our algorithm concerns the selection of basis degrees. Based on the structure of the HJB PDE (4), the initial condition $V(\cdot, 0)$ and the terminal negative log-density $V(\cdot, T) \approx -\log \mathcal{N}(0, I)$ generally require different spatial discretization spaces. Moreover, for each $t \in (0, T)$, the function $V(\cdot, t)$ is expected to remain sufficiently smooth, suggesting that only moderately sized, time-adapted spatial discretizations are required.

This adaptive choice of spatial discretization provides two main advantages:

- *Increased learning speed*: Reducing the number of degrees of freedom in each xTT representation decreases the computational cost of the minimization step, e.g. in solving (15).

- *Improved stability*: A reduced basis dimension lowers the sample complexity and improves the conditioning of the local least-squares problems. Moreover, numerical noise in unnecessary basis coefficients is suppressed, preventing spurious error propagation.

The idea is based on a simple heuristic. Fix a sparsity-gap parameter $s \in \mathbb{N}$ (typically $s = 1, 2$) and a loss tolerance $\text{tol}_{\text{loss}}$. At iteration step $k$, we first optimize the loss for a given xTT representation with rank $\boldsymbol{r}^{(0)}$ and basis degree vector $\boldsymbol{m}^{(0)} = (m_1^{(0)}, \ldots, m_d^{(0)})$.

- *Basis degree reduction:* Assume that the optimized loss $\mathcal{L}_{\min}^{(0)}$, obtained by solving (15), is below the tolerance $\text{tol}_{\text{loss}}$. Then, we reduce $\boldsymbol{m}^{(0)}$ as much as possible to obtain a new vector $\boldsymbol{m}^{(1)} \leq \boldsymbol{m}^{(0)}$, such that the corresponding minimized loss $\mathcal{L}_{\min}^{(1)}$ (computed using basis degrees $\boldsymbol{m}^{(1)}$) remains below $\text{tol}_{\text{loss}}$. We then set $\boldsymbol{m}^{(1)}$ as the new basis degree and proceed to the next iteration $k + 1$.

- *Basis degree increase:* Fix a maximum number of refinement steps $L \in \mathbb{N}$. If the optimized loss $\mathcal{L}_{\min}^{(0)}$, obtained from (15), exceeds the tolerance $\text{tol}_{\text{loss}}$, we increase $\boldsymbol{m}^{(0)}$ component-wise, yielding a sequence $\boldsymbol{m}^{(\ell+1)} \geq \boldsymbol{m}^{(\ell)}$ for

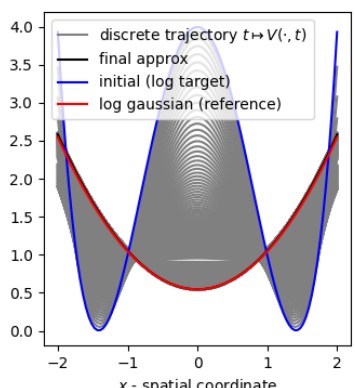 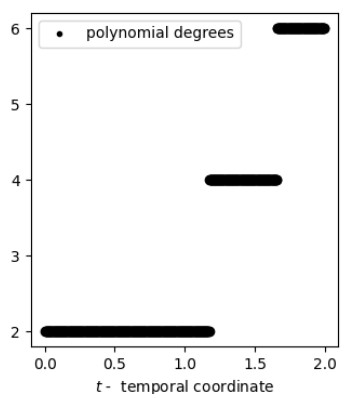 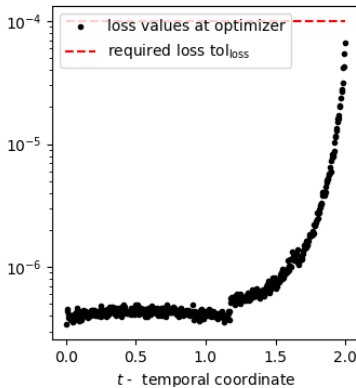

*Figure 10.* Illustration of basis-degree adaptivity for a one-dimensional example. Left: Trajectory transformation from the log-target distribution (log-multimodal) to the log-prior distribution (log-Gaussian). Middle: The selected polynomial degree decreases when moving from right ($t = T = 2$) to left ($t = 0$). In the limit $t \to 0$, the solution approaches a log-Gaussian, which requires only a degree of 2. Legendre polynomials are used as basis functions in this experiment. Right: Loss evolution and corresponding loss tolerance.

$\ell = 0, 1, \ldots, L$. This process is continued until either the minimized loss $\mathcal{L}_{\min}^{(\ell)}$ falls below the tolerance or no sufficient improvement in the loss reduction is observed over the last $s$ steps. In the latter case, we trigger rank adaptivity as discussed in Remark A.4.

*Remark* A.4. The sparsity gap becomes relevant when the chosen basis contains components that do not contribute meaningfully to the representation of the target function. For example, consider the function $f(x) = L_2(x) + L_4(x)$, where $L_i$ denotes the Legendre polynomials orthonormal in $L^2(-1, 1)$. Suppose we approximate $f$ using the basis $\{L_0, L_1, L_2\}$. In this case, enlarging the basis by adding $L_3$ – and thereby increasing the polynomial degree – does not reduce the projection error, since $L_3$ has no overlap with the nonzero components of $f$. More generally, when increasing the basis degree no longer yields a substantial reduction in the loss, it becomes preferable to increase the xTT rank, as described in Algorithm 4.

To illustrate the principle underlying our degree-adaptivity strategy, we consider a one-dimensional example, i.e. the multi-modal setup from Section 4.1 with $d = 1$. In this setting, there are no rank interactions, which allows us to isolate the effect of basis-degree adaptivity. Figure 10 illustrates this behavior in practice: for $T = 2$, the algorithm selects a polynomial degree of 6 in the initial optimization step of the BSDE loss, after which the degree is gradually reduced as $t \to 0$.

## A.7 Rank analysis

### A.7.1 EXPLICIT FUNCTIONAL TENSOR TRAIN RANKS

This section is devoted to the analysis of explicit FTT ranks for solutions of the HJB equation. It is well known that the Ornstein-Uhlenbeck process with a standard Gaussian invariant distribution, when initialized with a general Gaussian distribution, remains Gaussian throughout its evolution. Consequently, the associated HJB equation (see Lemma 2.1) admits, for every $t$, a solution $V(\cdot, t)$ that is a log-Gaussian potential.

For this reason, let us consider general Gaussian potentials of the form

$$\boldsymbol{x} \mapsto \boldsymbol{x}^\top M \boldsymbol{x}, \tag{111}$$

where $M$ is a symmetric and positive definite matrix.

It turns out that we can explicitly state the FTT ranks of these potentials, summarized in Theorem A.5 and Corollary A.6. This discussion is inspired by Gruhlke et al. (2026).

At the level of the potential, low-rank subdiagonal blocks have a very clear implication for the FTT ranks. We define the $i$-th subdiagonal block of $M$, for $i \in \{1, \ldots, d-1\}$, as the matrix $M_{1:i,i+1:d} \in \mathbb{R}^{i,(d-i)}$ in the following block decomposition:

$$M = \begin{bmatrix} M_{1:i,1:i} & M_{i+1:d,1:i} \\ M_{1:i,i+1:d} & M_{i+1:d,i+1:d} \end{bmatrix}. \tag{112}$$

As the following theorem shows, the ranks of the subdiagonal blocks of $M$ fully determine the FTT rank of $\boldsymbol{x}^\top M \boldsymbol{x}$.

**Theorem A.5** (FTT rank bounds for Gaussians with low-rank subdiagonal blocks). *Let $d \in \mathbb{N}$ and $\Phi \colon \mathbb{R}^d \to \mathbb{R}$ admit the form $\Phi(\boldsymbol{x}) = \boldsymbol{x}^\top M \boldsymbol{x}$ for a symmetric invertible matrix $M \in \mathbb{R}^{d,d}$. Furthermore, assume $M$ has sub-diagonal blocks $M_{1:i,i+1:d}$, $i = 1, \ldots, d-1$ with ranks given by $\ell_i \in \mathbb{N}$. Then $f$ has finite FTT rank $\boldsymbol{r} \in \mathbb{N}^{d-1}$. For $d \geq 3$,*

$$\boldsymbol{r} = 2 + (\ell_1, \ell_2, \ldots, \ell_{d-1}) \tag{113}$$

*and $\boldsymbol{r} = 2 \in \mathbb{N}$ for $d = 2$. In particular, in the case of an isotropic Gaussian, we have $\boldsymbol{r} \equiv 2$.*

*Proof.* First, note that there is only something to prove if the rank bounds $\ell_i$ satisfy

$$
\begin{aligned}
\ell_i \leq i + 2, &\qquad \text{if } i \leq \lfloor \tfrac{d}{2} \rfloor, \\
\ell_i \leq d - i + 2, &\qquad \text{if } i > \lfloor \tfrac{d}{2} \rfloor,
\end{aligned}
\tag{114}
$$

as the TT rank bounds will be higher than the maximal ranks otherwise. So let $i \in \{1, \ldots, d-1\}$ and $\ell_i$ satisfying the respective condition be the rank of the subdiagonal block $M_{i+1:d,1:i}$, which is defined by

$$M_{i+1:d,1:i} = \begin{pmatrix} m_{i+1,1} & \cdots & m_{i+1,i} \\ \vdots & & \vdots \\ m_{d,1} & \cdots & m_{d,i} \end{pmatrix}. \tag{115}$$

We consider a singular value decomposition of $M_{i+1:d,1:i}$ into

$$M_{i+1:d,1:i} = \underbrace{U}_{\in \mathbb{R}^{*,\ell_i}} \underbrace{\Sigma}_{\in \mathbb{R}^{\ell_i,\ell_i}} \underbrace{V^\top}_{\in \mathbb{R}^{\ell_i,*}}. \tag{116}$$

By construction, all terms in $\boldsymbol{x}^\top M \boldsymbol{x}$, including mixed interactions between variables indexed in $\{1, \ldots, i\}$ and those indexed in $\{i+1, \ldots, d\}$, are given by

$$2 \begin{pmatrix} x_{i+1} & \cdots & x_d \end{pmatrix} U \Sigma V^\top \begin{pmatrix} x_1 \\ \vdots \\ x_i \end{pmatrix} = 2 \left[ U^\top \begin{pmatrix} x_{i+1} \\ \vdots \\ x_d \end{pmatrix} \right]^\top \Sigma V^\top \begin{pmatrix} x_1 \\ \vdots \\ x_i \end{pmatrix}. \tag{117}$$

Hence, we have

$$
\begin{aligned}
f(x) &= \begin{pmatrix} 1 & 2x_{1:i}^\top & x_{1:i}^\top \cdot M_{1:i,1:i} \cdot x_{1:i} \end{pmatrix} \begin{pmatrix} x_{i+1:d}^\top \cdot M_{i+1:d,i+1:d} \cdot x_{i+1:d} \\ M_{1:i,i+1:d} \cdot x_{i+1:d} \\ 1 \end{pmatrix} \\[2mm]
&= \begin{pmatrix} 1 & 2x_{1:i}^\top & x_{1:i}^\top \cdot M_{1:i,1:i} \cdot x_{1:i} \end{pmatrix} \begin{pmatrix} 1 & 0 & 0 \\ 0 & M_{1:i,i+1:d} & 0 \\ 0 & 0 & 1 \end{pmatrix} \begin{pmatrix} x_{i+1:d}^\top \cdot M_{i+1:d,i+1:d} \cdot x_{i+1:d} \\ x_{i+1:d} \\ 1 \end{pmatrix} \\[2mm]
&= \begin{pmatrix} 1 & 2x_{1:i}^\top & x_{1:i}^\top \cdot M_{1:i,1:i} \cdot x_{1:i} \end{pmatrix} \begin{pmatrix} 1 & 0 & 0 \\ 0 & V\Sigma U^\top & 0 \\ 0 & 0 & 1 \end{pmatrix} \begin{pmatrix} x_{i+1:d}^\top \cdot M_{i+1:d,i+1:d} \cdot x_{i+1:d} \\ x_{i+1:d} \\ 1 \end{pmatrix} \\[2mm]
&= \begin{pmatrix} 1 & 2x_{1:i}^\top \cdot V & x_{1:i}^\top \cdot M_{1:i,1:i} \cdot x_{1:i} \end{pmatrix} \begin{pmatrix} 1 & 0 & 0 \\ 0 & \Sigma & 0 \\ 0 & 0 & 1 \end{pmatrix} \begin{pmatrix} x_{i+1:d}^\top \cdot M_{i+1:d,i+1:d} \cdot x_{i+1:d} \\ U^\top \cdot x_{i+1:d} \\ 1 \end{pmatrix} \\[2mm]
&= \underbrace{\begin{pmatrix} 1 & 2x_{1:i}^\top \cdot V & x_{1:i}^\top \cdot M_{1:i,1:i} \cdot x_{1:i} \end{pmatrix}}_{\in \mathbb{R}^{1,(\ell_i+2)}} \underbrace{\begin{pmatrix} x_{i+1:d}^\top \cdot M_{i+1:d,i+1:d} \cdot x_{i+1:d} \\ \Sigma U^\top \cdot x_{i+1:d} \\ 1 \end{pmatrix}}_{\in \mathbb{R}^{(\ell_i+2),1}}.
\end{aligned}
\tag{118}
$$

Since we have completely separated the first $i$ variables from the last $d - i$, this proves that the $i$-th rank is bounded by $\ell_i + 2$. Conversely, the rank cannot be lower than $\ell_i + 2$ as that would be in violation to $M_{i+1:d,1:i}$ having rank $\ell_i$. $\qquad \square$

The above result yields a worst case estimate of the ranks in the case that no structure of the precision matrix $M$ is known.

**Corollary A.6** (FTT rank bound for Gaussian potentials: worst case)**.** *Let* $d \in \mathbb{N}$ *and* $\Phi\colon \mathbb{R}^d \to \mathbb{R}$ *admit the form* $\Phi(\boldsymbol{x}) = \boldsymbol{x}^\top M \boldsymbol{x}$ *for a symmetric invertible matrix* $M \in \mathbb{R}^{d,d}$*. Then* $\Phi$ *has finite FTT rank* $\boldsymbol{r} \in \mathbb{N}^{d-1}$*. In particular for* $d \geq 3$,

$$\boldsymbol{r} \leq \overline{\boldsymbol{r}} := 2 + \begin{cases} \left(1, 2, \ldots, \frac{d}{2}, \ldots, 2, 1\right), & d \text{ even}, \\ \left(1, 2, \ldots, \frac{d-1}{2}, \frac{d-1}{2}, \ldots, 2, 1\right), & d \text{ odd}, \end{cases} \tag{119}$$

*and* $\boldsymbol{r} = 2 \in \mathbb{N}$ *for* $d = 2$.

*Proof.* Direct consequence of the sub-diagonal rank bounds $\ell_i$ of $M$ by application of Theorem A.5. $\qquad\square$

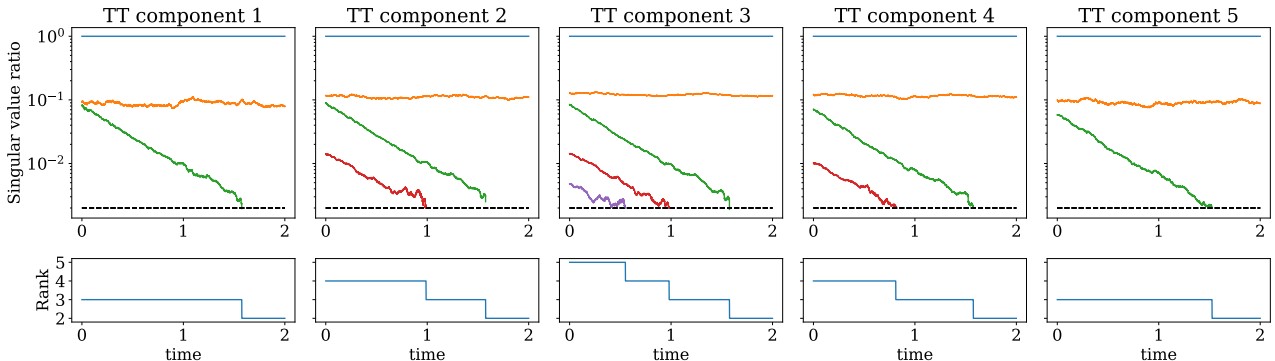

*Figure 11.* We display the singular values of all tensor train components for an example in dimension $d = 6$ over time and reduce the rank whenever a corresponding value is below a prespecified threshold.

### A.7.2   LOW-RANK AND SMOOTHNESS

This section is devoted to building intuition for the relationship between rank behavior and smoothness (or regularity). To this end, we distinguish between two cases for the FTT representation of a function $f$:

(1)  $f$ admits an $\varepsilon$-approximation by an FTT representation with finite rank $\boldsymbol{r}(\varepsilon)$,

(2)  $f$ admits an exact FTT representation with finite rank $\boldsymbol{r}$.

In addition to the discussion in Appendix A.7.1, we refer, for the second case, to Oseledets (2013), where explicit FTT ranks are derived for various classes of functions. In the present section, we focus on the first case, namely the approximation of functions by finite-rank FTT representations.

For simplicity, let $f \in L^2(D)$ with $D = [0,1]^d$, where $D$ serves as a prototype compact tensor domain and $L^2(D)$ denotes the classical Lebesgue space of square-integrable (equivalence classes of) Lebesgue-measurable functions. Let

$$\{p_0, p_1, \ldots\} \subset L^2(0,1) \tag{120}$$

be an orthonormal polynomial basis of $L^2(0,1)$. Its tensor-product construction then yields an orthonormal basis of $L^2(D)$. Expanding $f$ in this basis gives

$$f(\boldsymbol{x}) = \sum_{\nu \in \mathbb{N}_0^d} c_\nu \prod_{i=1}^d p_{\nu_i}(x_i). \tag{121}$$

Additional smoothness assumptions on $f$ imply asymptotic decay properties of the coefficients $c_\nu$. Such decay estimates can subsequently be used to derive bounds on the FTT rank

$$\boldsymbol{r} = (r_1, \ldots, r_{d-1}), \tag{122}$$

through the TT ranks of suitably truncated coefficient tensors $c_{\nu,\varepsilon}$.

We briefly summarize two smoothness classes that are particularly prominent in the theory of random PDEs: functions belonging to anisotropic mixed Sobolev spaces, following Griebel et al. (2023), and functions admitting holomorphic extensions.

**Anisotropic mixed Sobolev smoothness.** Assume that $f \in \bigotimes_{i=1}^{d} H^{s_i}(0,1)$ with $0 < s_1 \leq \cdots \leq s_d$. Then the expansion coefficients satisfy

$$|c_\nu| \lesssim \prod_{i=1}^{d} (1 + \nu_i)^{-s_i}. \tag{123}$$

Defining $\beta_1 := s_1 + s_2$ and $\beta_i := s_i$ for $i \geq 2$, there exists an FTT approximation $f_{\mathrm{FTT}}$ such that

$$r_i = \lceil \varepsilon^{-1/\beta_i} \rceil \qquad \text{and} \qquad \|f - f_{\mathrm{FTT}}\|_{L^2(D)} \lesssim \sqrt{d}\,\varepsilon. \tag{124}$$

In particular, increasing smoothness leads to smaller admissible ranks. More precisely, as $s_i$ increases, the corresponding ranks $r_i$ may stabilize, in the sense that $r_i \to 1$ as $\beta_i \to \infty$.

**Holomorphic extension.** Assume that $f$ admits a holomorphic extension with polyradii $1 < \rho_1 \leq \cdots \leq \rho_d$. In this case, one obtains the bound

$$|c_\nu| \lesssim \prod_{i=1}^{d} \rho_i^{-\nu_i}. \tag{125}$$

Sparse approximation then suggests truncation at $\bar{r}_i \sim \frac{\ln(1/\varepsilon)}{\ln \rho_i}$, yielding the truncated index set

$$\Lambda = \{\nu \in \mathbb{N}_0^d \mid \nu_i \leq \bar{r}_i\} \tag{126}$$

for the sparse polynomial approximation based on the truncated coefficient tensor $[c_\nu]_{\nu \in \Lambda}$.

We consider the unfolding obtained by merging the first $k$ dimensions and the last $d-k$ dimensions, i.e. the $k$-th matricization of $[c_\nu]_{\nu \in \Lambda}$. Using the inequality $1 + t \leq e^t$, the matrix rank of this unfolding yields the coarse bound

$$r_k \leq \prod_{i=1}^{k} (\bar{r}_i + 1) \lesssim \exp\left(\ln(1/\varepsilon) \sum_{i=1}^{k} \frac{1}{\ln(\rho_i)}\right). \tag{127}$$

Hence, the rank growth is governed by the summability properties of the sequence $(\ln(\rho_i)^{-1})_{i=1}^{d}$.

Assume that $\rho_i \sim e^{ai^b}$ for $a, b > 0$. Then $\ln(\rho_i) \sim ai^b$, and therefore

$$\sum_{i=1}^{k} \frac{1}{\ln(\rho_i)} \sim \frac{1}{a} \sum_{i=1}^{k} i^{-b}. \tag{128}$$

This yields, for $C = 1/a$,

$$r_k \lesssim \exp\left(C \ln(1/\varepsilon) \sum_{i=1}^{k} i^{-b}\right). \tag{129}$$

Summarizing, we obtain the following growth behaviour with respect to $k$:

$$r_k \lesssim \begin{cases} \exp(Ck^{1-b}), & 0 < b < 1 \quad \text{sub-exponential growth,} \\ k^{C \ln(1/\varepsilon)}, & b = 1 \quad \text{polynomial growth,} \\ \varepsilon^{-C}, & b > 1 \quad \text{uniformly bounded,} \end{cases} \tag{130}$$

where in the case $b > 1$ we absorb the finite value of $\sum_{i=1}^{\infty} i^{-b}$ into the constant.

As a final remark, we point out that smoothness is not necessary in order to obtain low-rank structure. In fact, product functions of the form

$$f(\boldsymbol{x}) = \prod_{i=1}^{d} f_i(x_i),\tag{131}$$

with each $f_i \in L^2(0,1)$, have minimal possible FTT rank

$$\boldsymbol{r} = (1, \ldots, 1) \in \mathbb{N}^{d-1}.\tag{132}$$

## B   Numerical details

### B.1   Limitations

While the outer iterations of our algorithm encourage the discovery of new modes (see Figure 1), we note that there is no guarantee that all modes of the target distribution will be identified – although this limitation is shared by all existing sampling algorithms. A further restriction arises from the requirement that the value function $V$ admits a sufficiently low-rank representation, which may not hold for all target distributions. Finally, although we have substantially improved the stability of a naive implementation of Algorithm 1 combined with FTTs (see Remark 3.3 and our experimental results), certain target distributions may still pose additional challenges, particularly due to the expressive and numerical limitations of FTTs.

### B.2   Evaluation metrics

In this section, we define the evaluation metrics used to measure sampling quality. Our metrics are based on path weights

$$w = \frac{\mathrm{d}\mathbb{P}_{\bar{Y}}}{\mathrm{d}\mathbb{P}_{X^u}}(X^u) \approx \frac{p_{\text{target}}(\widehat{X}_N^u) \prod_{n=0}^{N-1} \bar{p}_{n|n+1}(\widehat{X}_n^u | \widehat{X}_{n+1}^u)}{p_{\text{prior}}(\widehat{X}_0^u) \prod_{n=0}^{N-1} \vec{p}_{n+1|n}(\widehat{X}_{n+1}^u | \widehat{X}_n^u)},\tag{133}$$

where $\widehat{X}^u$ is the discretized SDE, as defined in (8) (Richter & Berner, 2024; Blessing et al., 2024). Here, $\vec{p}_{n+1|n}$ and $\bar{p}_{n|n+1}$ are the forward and backward transition densities of the discrete forward and backward process, respectively, given by

$$\bar{p}_{n|n+1}(\widehat{X}_n^u | \widehat{X}_{n+1}^u) = \mathcal{N}\left(\widehat{X}_n | \widehat{X}_{n+1} - f(\widehat{X}_{n+1}, (n+1)\Delta t)\Delta t, \sigma^2((n+1)\Delta t)\Delta t\right)\tag{134}$$

$$\vec{p}_{n+1|n}(\widehat{X}_{n+1}^u | \widehat{X}_n^u) = \mathcal{N}\left(\widehat{X}_{n+1} | \widehat{X}_n + (f + \sigma u)(\widehat{X}_n, n\Delta t)\Delta t, \sigma^2(n\Delta t)\Delta t\right).\tag{135}$$

Since in practice we typically can only evaluate $p_{\text{target}}$ up to the normalizing constant $\mathcal{Z}$, we can replace it by its unnormalized version $\rho_{\text{target}}$, bringing us unnormalized (discretized) importance weights, which we call $\widehat{w}$. We can now define the (normalized) effective sampling size (ESS) as

$$\text{ESS} := \frac{\left(\sum_{k=1}^{K} \widehat{w}^{(k)}\right)^2}{K \sum_{k=1}^{K} \left(\widehat{w}^{(k)}\right)^2},\tag{136}$$

which ranges from 0 to 1, with 1 being optimal. Here and in the sequel $K$ denotes the sample size. Further, we can define the log-variance divergence as

$$D_{\text{LV}} := \text{Var}(\log w) \approx \frac{1}{K-1} \sum_{k=1}^{K} \left(\log \widehat{w}^{(k)} - \frac{1}{K} \sum_{k=1}^{K} \log \widehat{w}^{(k)}\right)^2,\tag{137}$$

whose optimal value is zero (Richter et al., 2020; Nüsken & Richter, 2021). Finally, the normalizing constant $\mathcal{Z}$ can be computed via

$$\mathcal{Z} = \mathbb{E}[w] \approx \frac{1}{K} \sum_{k=1}^{K} \widehat{w}^{(k)}.\tag{138}$$

## B.3 Hyperparameter dependence

In this section, we empirically examine how selected hyperparameters influence the performance of our FTT training algorithm. We focus on the Multiwell problem introduced in Section 4 and consider the setting $d = 10$, $\delta = 2$, with $m \in \{3, 10\}$, corresponding to $2^3$ and $2^{10}$ modes, respectively. We investigate the effect of varying the batch size, the number of basis functions, and the domain shrinking factor $p$ (see Appendix A.4.4), as illustrated in Figure 12 and Figure 13. Unless stated otherwise, we fix the number of basis functions to 10, the batch size to $K = 2^{15}$, and set $p = 0.1$.

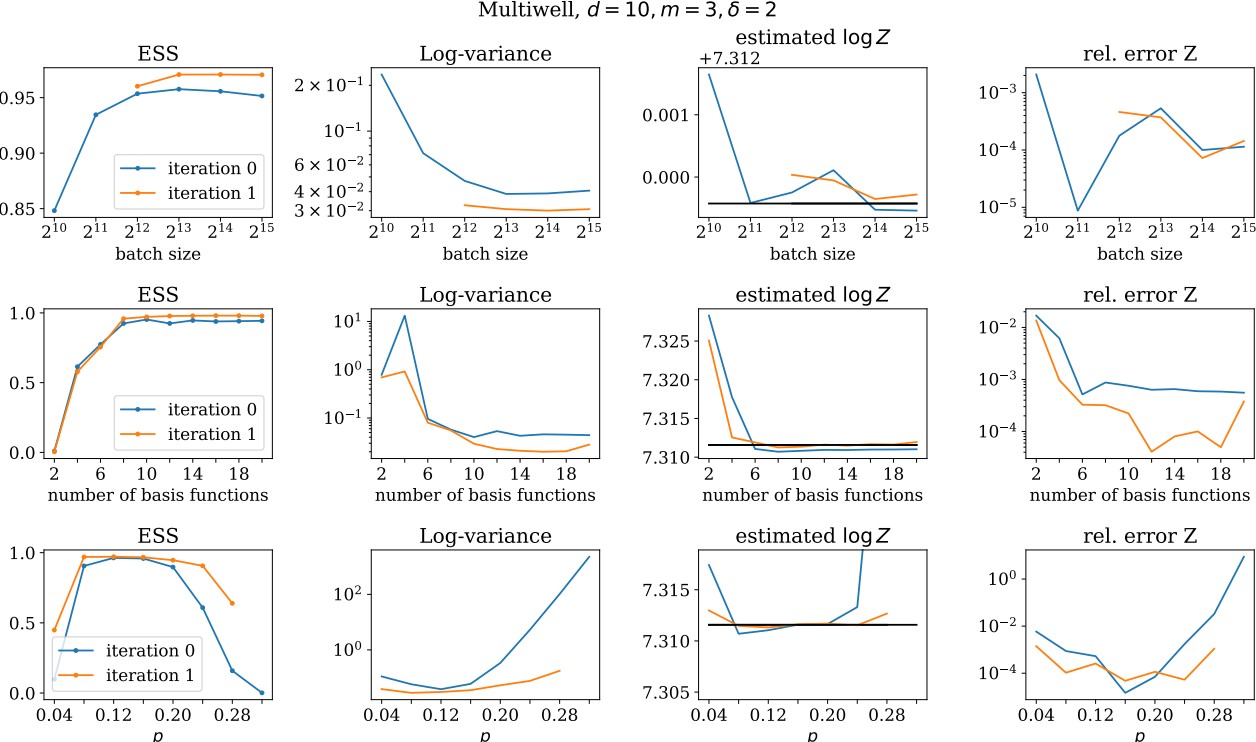

*Figure 12.* We assess the sensitivity of our algorithm by varying key hyperparameters and recording the effective sample size (ESS), the log-variance divergence, the log-normalizing constant, and the relative error in estimating the normalizing constant. Whenever a value is absent from a plot, the algorithm diverged for that configuration. The results indicate that performance improves with increasing batch size, but the gains tend to saturate beyond a certain threshold. A similar trend is observed for the number of basis functions. In contrast, the parameter $p$ exhibits a clear optimal range: values that are too small or too large degrade performance.

## B.4 Comparison with alternative sampling methods

In Figure 14, we compare the performance of our TTD sampler against various baselines, covering state-of-the-art neural samplers based on ODEs and SDEs and combinations with MCMC methods. For all baselines we leverage the public repository by Chen et al. (2024) (which is derived from the repository by Blessing et al. (2024)). We refer to these works for details on each baseline. In particular, we use the default hyperparameters and tune the scale of the prior and diffusion coefficient if applicable. We report the error in estimating $\log Z$ and the ESS (if available for the given baseline) and observe that our TTD sampler consistently achieves better performance for a given time budget as well as better final performance than all considered baselines.

Further, adding to the experiments in Section 4.1, we additionally compare our TTD sampler with the PIS sampler in more detail (Zhang & Chen, 2022), a well-established diffusion-based sampling method. Note, however, that the underlying PDE is different from our HJB PDE stated in Lemma 2.1. Further, we want to highlight that PIS has a natural advantage when considering the Multiwell problem defined in (18) since its initialization samples from a Gaussian density. In Figure 15 we repeat the same plot as in Figure 4, now integrating PIS as well.

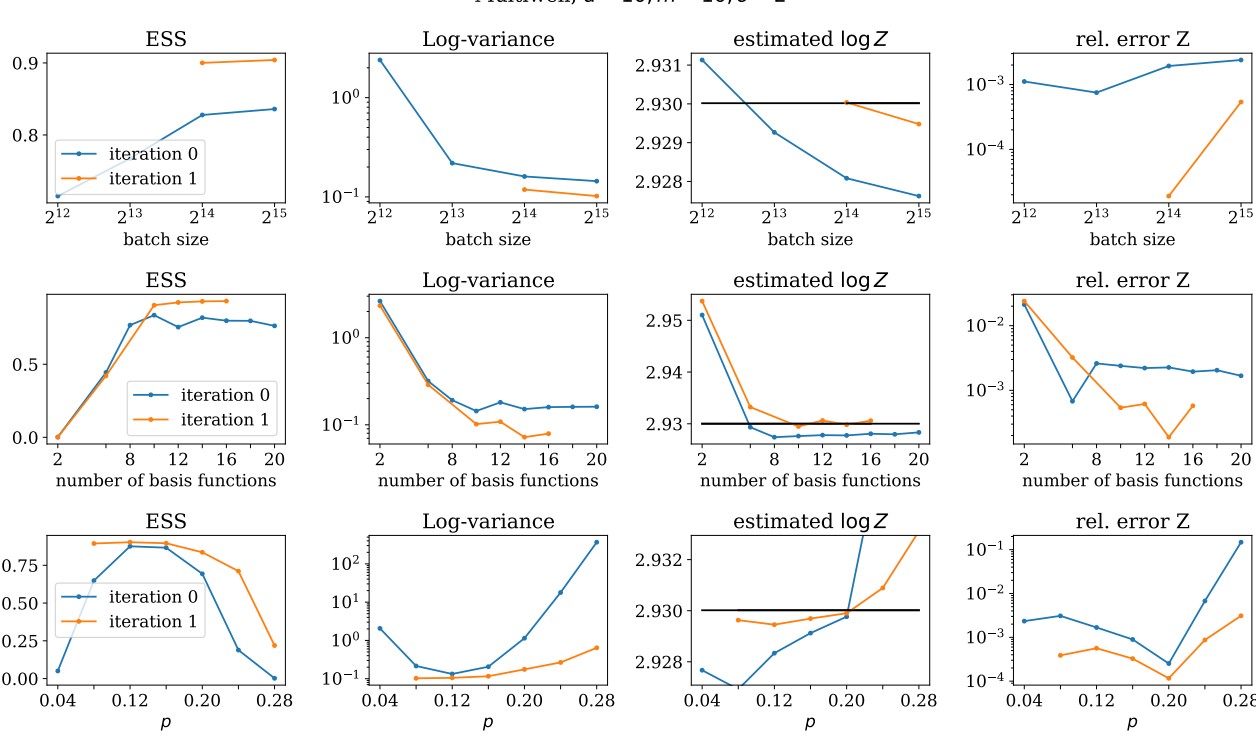

*Figure 13.* As in Figure 12, we assess the sensitivity of our algorithm by varying key hyperparameters, now on a Multiwell problem with $2^{10}$ modes. We observe the same qualitative trends as before; however, in this highly multimodal setting a larger batch size and a greater number of basis functions are required to stabilize the algorithm.

### B.5  Additional experiment: Kitagawa nonlinear state space model

We evaluate our method using a modified version of the Kitagawa nonlinear state space model, designed to isolate high-dimensional geometric complexity while maintaining a unimodal posterior (Kitagawa, 1996). The latent state evolves according to nonlinear dynamics defined by

$$x_n = \frac{1}{2}x_{n-1} + \gamma \frac{x_{n-1}}{1 + x_{n-1}^2} + v_n \tag{139}$$

where $\gamma$ controls the nonlinearity strength and $v_n$ represents Gaussian process noise. Unlike the standard benchmark, we employ a linear observation model to break the symmetry inherent in the original formulation, namely

$$y_n = x_n + w_n. \tag{140}$$

This modification eliminates multi-modality but preserves the challenging correlations in the joint posterior, providing a robust test for sampling efficiency on high-curvature manifolds. Given the fixed initial state $x_0 = 0$ and the observed sequence $y_{1:M}$, the joint log-posterior density $\log \rho(x_{1:M})$ is given by

$$\log \rho_{\text{target}}(x_{1:M} \mid y_{1:M}) = -\sum_{n=1}^{M} \left[ \frac{1}{2\sigma_v^2} \left( x_n - f(x_{n-1}) \right)^2 + \frac{1}{2\sigma_w^2} \left( y_n - x_n \right)^2 \right], \tag{141}$$

where $f(x_{n-1})$ represents the nonlinear transition mean

$$f(x_{n-1}) = \frac{1}{2}x_{n-1} + \gamma \frac{x_{n-1}}{1 + x_{n-1}^2}. \tag{142}$$

In our experiments we choose $\sigma_v = \sigma_w = 1$ in $d = 10$ and vary the nonlinear strength $\gamma$, see Table 2.

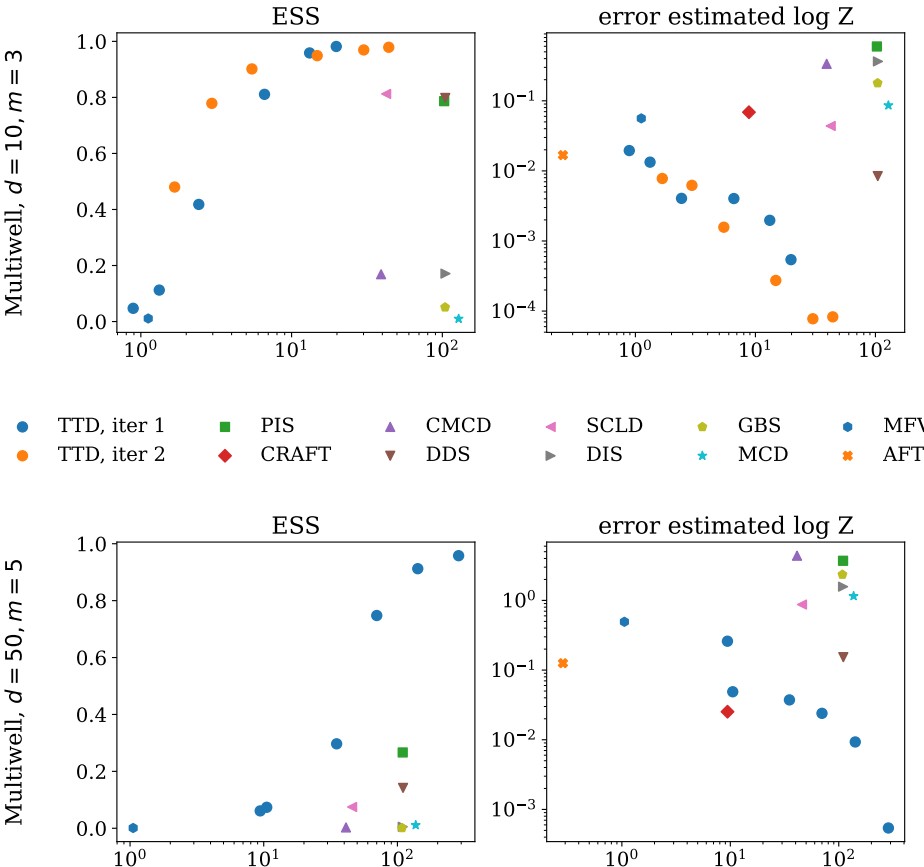

*Figure 14.* We compare the performance of our TTD sampler against several state-of-the-art baselines, including PIS (Zhang & Chen, 2022), DDS (Vargas et al., 2023), DIS (Richter & Berner, 2024), GBS (Blessing et al., 2024), CMCD (Vargas et al., 2024), SCLD (Chen et al., 2024), CRAFT (Matthews et al., 2022), AFT (Arbel et al., 2021), MFVI (Bishop, 2006), and MCD (Doucet et al., 2022). We see that TTD converges significantly faster and outperforms the baselines in terms of both estimation of normalizing constants as well as ESS (if available for the given baseline). We refer to Appendix B.4 for further details.

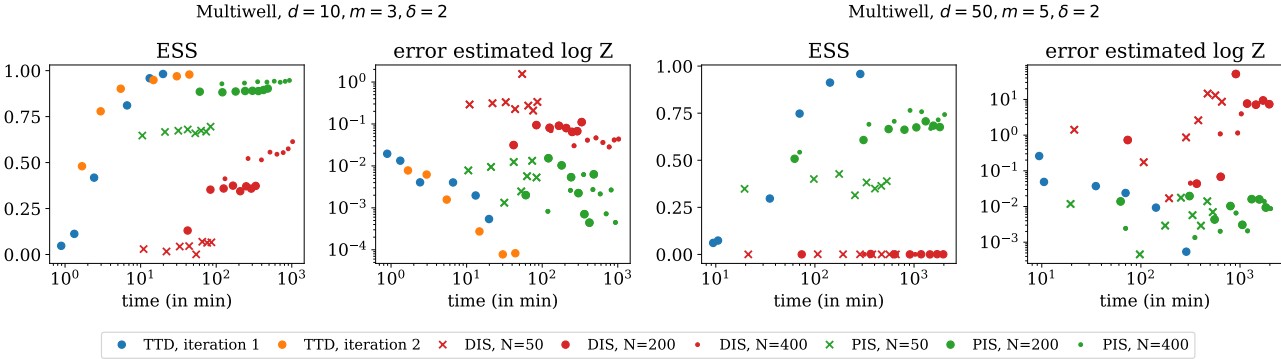

*Figure 15.* We compare the performance versus runtime of our TTD sampler with DIS (Berner et al., 2024) and PIS (Zhang & Chen, 2022). By design, our algorithm produces one result per chosen number of steps $N$ (shown as blue and orange dots), whereas DIS and PIS can improve over training time. Accordingly, we evaluate DIS and PIS at equally spaced runtime intervals. In both experiments, our algorithm is not only faster but also achieves better results, particularly in settings where DIS can exhibit instability.

*Table 2.* We display the effective sample size (ESS) and the log-variance divergence for the Kitagawa nonlinear state space model for varying nonlinear strength $\gamma$ and different amounts of steps $N$.

| $\gamma$ | $N$ | ESS | Log-variance |
|---|---|---|---|
| 0.5 | 2048 | 0.925 | 0.076 |
| | 4096 | 0.937 | 0.068 |
| 1.0 | 2048 | 0.897 | 0.237 |
| | 4096 | 0.911 | 0.133 |
| 1.5 | 2048 | 0.885 | 0.280 |
| | 4096 | 0.898 | 0.192 |
| 2.0 | 2048 | 0.858 | 0.970 |
| | 4096 | 0.871 | 0.612 |

