# OpenReview forum: "Tensor Train Diffusion: Leveraging Low-Rank Structures for High-Dimensional Score-Based Sampling"
_ICML.cc/2026/Conference — ICML 2026 regular_

### Official Review · Reviewer_LEB8 · 2026-02-14

**Soundness:** 3
**Presentation:** 2
**Significance:** 3
**Originality:** 4
**Overall Recommendation:** 5
**Confidence:** 2

**Summary:**

This papers provides a method to learn score functions when we cannot directly access the data. Its extends from a PDE constraint of the score function. In the grid setting, this PDE residual can be written into a regression loss. To further make the training efficient, FFT is applied with ALS. Experiments on synthetic data validate the effectiveness of the method.

**Compliance With Llm Reviewing Policy:**

Affirmed.

**Final Justification:**

My concerns have been addressed.

**Key Questions For Authors:**

See weakness.

Additional:

I have read serveral paper to train diffusion model by model a PDE target. But to be honest here, I do not quite understand why we need to assume that we cannot access the data. I hope the author can explain this to me, or I think it wrongly.

**Limitations:**

yes

**Strengths And Weaknesses:**

Strength:
1. The paper is noval and theortical sound. I introduce a new training method of score model and apply serveral neccessy modules, such as FFT and ALS. Experiments in both main text and appendix compare among serveral baselines and further validate the proposed method.

Weakness:
1. The author mention that the method suffers from error propagation. However, no mitigation method is proposed to the best of my knowledge.
2. FTT heres is introduced to improved the training efficiency. However, it is not reasoned why we can model V in this way. Does this mean that V has certain low rank structure?
3. It seems that the experiments only contain one synthetic data. Maybe this is not sufficient. Could author clarify this?

---

> ### Author Rebuttal · Authors · 2026-03-26
>
> Dear Reviewer LEB8. We thank you for your review and are happy that you value our contribution as novel and theoretical sound and that you appreciate our numerical validation.
>
> **Potential error propagation**
>
> While error propagation is theoretically possible, we did not observe it in our experiments as long as the number of time steps $N$ is sufficiently large (Figures 3 and 6). Increasing $N$ consistently improves performance.
>
> Error propagation could be further mitigated by jointly optimizing multiple time steps, which integrates naturally into our tensor-train and BSDE framework with minimal extra cost. However, this was not necessary to achieve strong performance in our experiments.
>
> **FTT modeling of $V$**
>
> Functional tensor trains (FTTs) are introduced to improve training efficiency, but they are not merely a computational convenience. Rather, they form a nonlinear (multi-linear) approximation class specifically designed for high-dimensional function approximation, with the goal of mitigating the curse of dimensionality. Alternative approximation paradigms include, for example, active subspace methods, sparsity-based approaches, or compositional structures such as neural networks. A key advantage of FTTs is that they admit mathematically well-structured optimization procedures, including stable least-squares formulations and Riemannian optimization methods, which explicitly exploit the geometry of the low-rank tensor manifold and enable efficient training.
>
> Regarding the modeling assumption: the use of an FTT ansatz for $V$ does implicitly rely on the expectation that $V$ exhibits some form of low-rank structure, at least approximately. In our setting, $V$ solves an HJB equation and represents trajectories of (negative) log-densities. When the underlying stochastic dynamics are driven, e.g., by an OU process, the resulting densities are smoothed through convolution with Gaussian kernels. Consequently, the corresponding log-densities inherit a significant degree of regularity. This regularity provides theoretical support for low-rank approximability. In particular, there is a well-established connection between smoothness properties (e.g., in Sobolev or Besov spaces, see [1, 2]) and compressibility in tensor formats. Closely related is the notion of sparsity: when functions admit sparse representations in tensor-product bases, this often translates directly into low-rank structure in TTs.
>
> From an application perspective, this assumption is further supported in areas such as Bayesian inversion and uncertainty quantification. There, the log-posterior density inherits structural properties from the underlying forward model, which is frequently governed by (random) partial differential equations. In many standard modeling approaches, such as those based on Karhunen-Loève expansions, the associated solution maps are smooth and exhibit low effective dimensionality. This, in turn, promotes sparse and low-rank representations of the log-posterior (see, e.g., [3]).
>
> In summary, while low-rank structure of $V$ is not guaranteed in full generality, there are strong theoretical and practical indications that such structure arises in many relevant settings, thereby justifying the use of FTTs as an efficient and well-founded approximation class. As a consequence, we plan to include a brief discussion of this aspect in the revised manuscript, space permitting.
>
> **Experiments only contain synthetic data**
>
> Thank you for raising this important point. Our framework relies on the assumption that the target distribution is specified through its unnormalized density, which serves as the terminal boundary condition of the PDE stated in Lemma 2.1. This is fundamentally different from the classical diffusion model setting, where learning is based on data samples rather than an explicit density. As a result, our work addresses the classical sampling problem, which is arguably more challenging than generative modeling, since no information about the location of high-probability regions (modes) is available through data. Consequently, our experiments necessarily consider settings where an unnormalized density is explicitly given. Examples include energy-based models from physics (Section 4.3, $\phi^4$ scalar field theory) and posterior densities arising in Bayesian statistics (Section B.5, Kitagawa nonlinear state-space model).
>
> We will emphasize this distinction more clearly in the revised version. Please let us know if further clarification would be helpful.
>
> **Why can we not access data from the target?**
>
> See comment above: In this setting, no data from the target distribution is available by definition since we study the classical sampling problem. We will clarify this distinction more explicitly in the revised version and are happy to provide further clarification if helpful.
>
>
> [1] Ali, M., & Nouy, A. (2023).
>
> [2] Griebel et al. (2023).
>
> [3] Bachmayr et al. (2018).
>
> (See response to Reviewer 1obN for reference details.)

---

> > ### Author Rebuttal · Reviewer_LEB8 · 2026-04-01
> >
> > I would like to thank authors for the replies. I now understand that the relevent field can be modeled by FTT is because their internal gaussian noise, which is interesting.
> >
> > Regarding the application scenario, I now understand they can be energy-based models from physics and posterior densities arising in Bayesian statistics. Sorry, I do not understand these application indeed, but thanks for the clarification!!!
> >
> > I believe this work is good, but I have just one more question. Why jointly optimizing multiple time steps only gives minimal extra cost in your modelling? It may need backpropagation through the entire time series?

---

> > > ### Author Response · Authors · 2026-04-01
> > >
> > > Dear Reviewer LEB8,
> > >
> > > Thank you for your response. We are happy to provide further clarification, in particular regarding the idea of optimizing multiple time steps jointly, which we could only briefly sketch previously due to space limitations.
> > >
> > > The algorithm presented in the paper relies on single time-step optimization, where, in an alternating fashion, only linear least-squares problems need to be solved. Importantly, regardless of how many time steps are optimized, we first simulate the forward SDE once to generate the samples that are then reused throughout the entire optimization cycle (backward in time). Hence, considering multiple time steps does not increase the sampling cost.
> > >
> > > At the other extreme, for a given time discretization, one could attempt to optimize all $N$ time steps simultaneously starting from a random surrogate model. In practice, this naive approach would typically be intractable. For this reason, we propose a middle-ground strategy:
> > >
> > > - **Prediction step:** Optimize $k < N$ time steps sequentially using the original algorithm.
> > > - **Correction step:** Jointly optimize the last $k$ time steps using the previously computed solutions as initialization.
> > > - Continue with the next block of time steps.
> > >
> > > Because this procedure starts from a well-informed initialization, the correction step amounts to updating a tensor-product structure of functional tensor trains (FTTs), which allows the use of efficient alternating or Riemannian optimization methods.
> > >
> > > For $k>1$, the corresponding loss depends nonlinearly on $k-1$ input FTTs, in contrast to the single-step loss, which is linear in the input FTT. However, due to the good initialization, only a small number of iterations is required even when optimizing multiple time steps jointly. We therefore expect only marginally longer runtimes.
> > >
> > > Please let us know if you have further questions. We would also be grateful if this clarification helps in reassessing your evaluation.

---

### Official Review · Reviewer_LFYY · 2026-02-22

**Soundness:** 3
**Presentation:** 2
**Significance:** 2
**Originality:** 4
**Overall Recommendation:** 4
**Confidence:** 3

**Summary:**

The paper introduces tensor train diffusion, a method for sampling from unnormalized probability densities by solving the HJB underlying diffusion-based samplers using functional tensor trains (FTT). The method combines the FTT representation with a backward-in-time iterative scheme derived from BSDEs, replacing SGD-based optimization with alternating least squares. The paper claims to consider a broad theme connecting diffusion models, HJB equations, BSDEs, and low-rank tensor approximation. The paper shows that tensor train representations can serve as efficient alternatives to neural networks for score approximation in diffusion-based sampling.

**Compliance With Llm Reviewing Policy:**

Affirmed.

**Key Questions For Authors:**

How does TTD perform when the target distribution has dense (non-sparse, non-banded) correlations, e.g., a Gaussian with a random dense precision matrix in d=50?

**Limitations:**

yes

**Strengths And Weaknesses:**

Strengths

The replacement of neural networks with FTTs for solving the HJB equation is creative and well-justified. The authors clearly articulate the shortcomings of neural PDE solvers and present a compelling alternative. The paper demonstrates substantial effort in making the method practical and robust, resulting in compelling empirical results in figure 3, 4, and 14 are compelling.

Weaknesses

While the experiments are well-executed, the target distributions are relatively structured: in Multiwell, the potential is separable in coordinates beyond index m, and the multimodal components are axis-aligned double wells. This structure is inherently favorable for tensor decomposition; in Ginzburg-Landau, only nearest-neighbor coupling, which implies banded interaction matrices and naturally low TT ranks; The Kitagawa modelis a chain-structured model, again, favorable for TT. It remains unclear how TTD would perform on targets with dense, long-range correlations or where the low-rank structure is absent or requires very high ranks. The paper acknowledges this in B.1, but a more explicit discussion or failed-case analysis would strengthen the contribution. The highest dimension tested is d=50 (Multiwell) and d=30 (Ginzburg-Landau). While the paper claims linear scaling in d for storage, the actual runtime grows from 10 to 300 minutes for d=50. The runtime scaling with dimension, rank, and basis size deserves more systematic study.

---

> ### Author Rebuttal · Authors · 2026-03-26
>
> Dear reviewer LFYY, we thank you for your careful and thorough review. We appreciate your recognition of our proposal to replace neural networks with FTT representations for score approximation in diffusion-based sampling, as well as your positive assessment of our empirical results. We address your questions and concerns in detail below.
>
>
> **The target distributions are relatively structured.**
>
> We thank the reviewer for this detailed and thoughtful feedback. Indeed, the examples of Multiwell, Ginzburg-Landau, and Kitagawa exhibit structure that can be favorable for FTT representations. However, these benchmarks are standard in the sampling literature and provide a well-understood testbed for validating both accuracy and computational efficiency.
>
> We agree that TT methods are more challenging for distributions with dense, long-range correlations, where ranks -- and thus storage and computation -- can grow significantly. This is a general limitation of low-rank representations, not our algorithm. Still, TT methods remain effective for moderately correlated or weakly entangled distributions, which often exhibit approximate low-rank structure in high-dimensional applications.
>
> Notably, in Section 4.2 we investigate a Gaussian target with a random full-rank covariance matrix. Here, the optimal TT rank can be computed analytically, and Figures 5 and 11 demonstrate that our rank-adaptive algorithm successfully discovers the correct rank during optimization (see Section A.6 in the appendix for details).
>
> As noted, we have already highlighted these limitations in Section B.1. If space permits in the final version, we plan to move this discussion to the main text to make it more prominent. Overall, we believe our approach demonstrates the feasibility and potential of TT-based sampling in high-dimensional problems with partially exploitable low-rank structure, while acknowledging its limitations in fully dense, high-rank settings.
>
> Finally, while the considered examples have structured target densities, this structure does not directly extend to the full HJB solution trajectory. The function $V$, as a solution of a nonlinear PDE, can develop more complex dependencies over time, even if the terminal density is simple or locally structured.
>
> As a result, both the effective ranks and features of $V$ may vary significantly along the trajectory before ultimately converging to a simple log-Gaussian potential (with low rank $\mathbf{r}=(2,\dots,2)$). This motivates our use of adaptive tensor ranks and basis degrees, allowing the approximation to adjust to the evolving complexity rather than relying solely on favorable structure in the target density (see Appendix A.6).
>
>
> **Scale dependence on dimension $d$**
>
> The overall computational cost is influenced by several additional factors, including the tensor ranks, the chosen basis sizes, and the cost of the underlying optimization routines (e.g., alternating least squares) as well as repeated function and gradient evaluations.
> In our experiments, the growth to 10-300 minutes for $d = 50$ reflects both modestly increased TT ranks and the need for finer basis discretizations to maintain accuracy.
>
>
> The storage of the $i$-th TT core scales as $r_{i-1} n_i r_i$. For ranks growing with dimension -- e.g., Gaussian log-densities in worst case with linear-in-$d/2$ ranks (cf. Corollary A.6) -- the central cores can have $\mathcal{O}((d/2)^2)$ entries. Consequently, tensor contractions and optimization scale polynomially in $d$.  While TT formats still offer large savings over full tensors, practical runtime depends on dimension, rank growth, and basis size.
>
> We agree that a more systematic investigation of these dependencies would be valuable, and we will expand the discussion in the revised manuscript to better clarify the observed scaling behavior in our experiments and to provide clearer guidance for practitioners.
>
>
> **TTD performance for Gaussian targets with full-rank covariance matrices**
>
> As noted previously, for a Gaussian target with a full-rank covariance matrix, the true optimal TT rank can be computed analytically (see Theorem A.5 and Corollary A.6 in Section A.7 of the appendix). Our numerical experiments, shown in Figures 5 and 11, demonstrate that our rank-adaptive algorithm successfully recovers these optimal ranks. In particular, as the Gaussian target converges toward a standard normal, the TT rank converges to 2, consistent with the theoretical prediction.

---

> > ### Author Rebuttal · Reviewer_LFYY · 2026-04-01
> >
> > I thank the authors for answering my questions. I do not have remaining questions, and will keep my score at weak accept.

---

> > > ### Author Response · Authors · 2026-04-03
> > >
> > > Dear Reviewer LFYY,
> > >
> > > Thank you again for taking the time to evaluate our paper. We are pleased that you value our contribution and support its acceptance.
> > >
> > > For your convenience, we briefly summarize the key contributions of our work below.
> > >
> > > * **A diffusion-based sampler without neural networks.**
> > > We introduce the first diffusion-based sampling algorithm that replaces neural networks (NNs) with functional tensor trains (FTTs). FTTs are particularly advantageous when the target distribution admits a low-rank structure (see our previous discussion).
> > >
> > > * **A BSDE-based, regression-driven training procedure.**
> > > By embedding the sampler in a BSDE framework, we leverage the structure of FTTs and develop a regression-based training algorithm. This avoids potentially slow SGD-based optimization and can be substantially faster. Although FTTs present their own numerical challenges, we address these through several algorithmic innovations, including appropriate basis constructions, outer-loop stabilization, dynamically moving domains along sample trajectories, adaptive regularization, and adaptive rank selection.
> > >
> > > * **Strong empirical speed-accuracy performance.**
> > > Our experiments show that the proposed sampler achieves a highly favorable speed-accuracy trade-off: it can be both faster and more accurate than existing (diffusion-based) samplers. In particular, it significantly improves upon DIS and PIS - even though it uses essentially the same SDE model - while requiring fewer target evaluations (see Figures 3, 14, 15).
> > >
> > > * **A first step toward FTT-based generative modeling.**
> > > We view this work as an initial step in bringing FTTs into diffusion-based sampling. Our results demonstrate that FTTs can yield faster and more robust algorithms, offering a promising alternative to the often fragile and slow optimization procedures required by neural networks.
> > >
> > > Please do not hesitate to reach out if you have any further questions.

---

### Official Review · Reviewer_1obN · 2026-03-11

**Soundness:** 3
**Presentation:** 4
**Significance:** 3
**Originality:** 2
**Overall Recommendation:** 4
**Confidence:** 3

**Summary:**

Summary: The main idea of the paper is quite simple. Lemma 2.1 shows the HJB PDE related to learning the score leading to Equation (7) as the loss in the variable $\widetilde{V}$. The authors then propose to solve it piecewise over time intervals leading to loss in Equation (9) and subsequently Equation (10). Algorithm 1 then follows naturally. The key idea is then introduced in section 3 where a low-TT rank approximation is proposed for functions $$\hat{V}_n$$, the in turn leads to the optimization problem in equation (15). Following that, as most papers on TT methods do, ALS is proposed in section 3.1 to estimate the tensor components from the data.
Section 4 presents simulations for densities that arise in MD problems and $\phi^4$ scalar field theory. An important aspect of the work is to choose the right tensor basis.

**Compliance With Llm Reviewing Policy:**

Affirmed.

**Key Questions For Authors:**

Questions:
1. If possible, test the method on simple distributions as well (mixtures of Gaussians) and some image distributions (MNIST). What would one expect in these cases?
2. What is the effect of time discretization (intervals) and is it necessary to have a uniform discretizaton? (I was not able to readily locate this ablation study, if indeed it is present in the paper).
3. How can one know whether to use this approach or the SGD based score-matching a priori?

**Limitations:**

Yes

**Strengths And Weaknesses:**

Strengths:
1. To the best of my knowledge, the approach presented is novel for training generative models.
2. The exposition is accessible and detailed and appendices really add the missing details with additional background and experiments.
3. The method outperforms baselines on the models considered.

Weakness:
1. The main and primary weakness is whether the method generalizes to non-scientific data such as images.
2. It is also not clear which densities will have a low-rank TT approximation in this context.
3. Score-matching and Diffusion models come with guarantees on generated data. It is not clear if the methods proposed here can have such guarantees

---

> ### Author Rebuttal · Authors · 2026-03-26
>
> Dear Reviewer 1obN, we thank you for your careful review. We are pleased that you recognize our contribution as novel and that you note that our method significantly outperforms alternative approaches.
>
> **Application of the method to non-scientific data (e.g., images).**
>
> This point reflects a misunderstanding of our setting. We do not assume access to data samples (e.g., images), but only to an unnormalized target density. Our work addresses the classical sampling problem arising in scientific and probabilistic modeling, where the distribution is given analytically rather than through data. This setting is arguably more challenging, as the locations of the modes are unknown a priori. (See also response to Reviewer ZXq6.)
>
> **Which densities admit a low-rank TT approximation**
>
> We do not assume low-rank structure of the densities themselves, but instead exploits the structure of the HJB solution, i.e., time-dependent trajectories of (negative) log-densities.
>
> This distinction is crucial: even simple densities such as Gaussians typically do not admit low-rank TT approximations [1], whereas their logarithms are significantly more structured. In particular, for log-Gaussian densities, TT ranks scale at most linearly in the dimension $d$ (cf. Corollary A.6).
>
> This observation is consistent with results from uncertainty quantification and Bayesian inversion, where solutions of random PDEs and consequently log-posteriors are known to admit low-rank or sparse representations (see, e.g., [2,3]).
>
> Overall, this supports the use of log-densities as a suitable and theoretically justified setting for low-rank TT approximations in many applications. Please let us know if you have additional questions and would like us to clarify further aspects.
>
> **Guarantees on generated samples.**
>
> Although we do not assume data samples but only an unnormalized density, the underlying SDE, time-reversal principle, and PDE are the same as in score-based generative modeling. Hence, existing theoretical guarantees directly apply: sampling quality is governed by the accuracy of the learned score function (see, e.g., [4]). Please let us know if you had a different aspect in mind.
>
> **Distributions, including images.**
>
> Our experiments cover diverse, challenging targets-high-dimensional Gaussian, multi-well potentials, $\phi^4$ scalar field theory, and the Kitagawa nonlinear state-space model-standard benchmarks for sampling from unnormalized densities. Image generation falls outside our scope: our method assumes access to the unnormalized density, not data samples, making it fundamentally different from data-driven generative modeling (see above + Reviewer ZXq6).
>
> **Effect of time discretization.**
>
> We studied the impact of the number of time steps $N$ on sampling performance, see Figures 3 and 6. Finer discretizations improve accuracy but increase computational cost, as analyzed in Figures 4 and 14. Non-uniform time grids could further optimize this trade-off, but we would like to leave this for future work, since uniform steps were sufficient in our experiments.
>
>
> **Our approach vs. SGD-based score matching (SM)?**
>
> Classical SM is not applicable here, since only an unnormalized density is available. Our TT approach works best when the log-target has a low-rank structure, enabling faster, more efficient sampling (Figures 4 and 14), while SGD-based methods may be preferable for high-rank or highly flexible targets.
>
> Low-rank structure is not incidental but can be theoretically justified. In particular, functions with sufficient smoothness (e.g., [3,5]) often admit low-rank approximations, as regularity and weak variable interactions promote compressibility in tensor formats. This is closely related to sparsity [2]: in tensor-product bases, sparse representations frequently correspond to low-rank structures.
>
> Overall, low-rank approximability is closely linked to properties such as smoothness, weak coupling, and low effective dimensionality - features commonly encountered in applications, especially in PDE-based models and Bayesian inference. While counterexamples exist, many practically relevant log-densities exhibit sufficient structure to make low-rank TT approximations effective.
>
> [1] Rohrbach, P. et al. (2022). Rank bounds for approximating gaussian densities in the tensor-train format. SIAM/ASA Journal on Uncertainty Quantification.
>
> [2] Bachmayr, M. et al. (2018). Parametric PDEs: sparse or low-rank approximations? IMA Journal of Numerical Analysis.
>
> [3] Griebel, M. et al. (2023). Low-rank approximation of continuous functions in Sobolev spaces with dominating mixed smoothness. Mathematics of Computation.
>
> [4] Chen, S. et al. (2022) Sampling is as easy as learning the score: theory for diffusion models with minimal data assumptions. In The Eleventh International Conference on Learning Representations.
>
> [5] Ali, M., et al. (2023). Approximation theory of tree tensor networks: Tensorized univariate functions. Constructive Approximation.

---

> > ### Author Rebuttal · Reviewer_1obN · 2026-04-03
> >
> > Most of the concerns have been addressed. However, I would still like the authors to show how existing guaranteed in reference [4] mentioned in the rebuttal -
> >
> > [4] Chen, S. et al. (2022) Sampling is as easy as learning the score: theory for diffusion models with minimal data assumptions. In The Eleventh International Conference on Learning Representations -
> >
> > apply to their case. My conjecture is that the errors will propagate in their scheme across the intervals.
> >
> > Further, I request authors to provide a simple example, even if taken from the papers referenced in the rebuttal, of low-rankness that arises from smoothness etc. A simple illustrative example can go a long way for the ML audience.

---

> > > ### Author Response · Authors · 2026-04-06
> > >
> > > We are glad our previous explanations were helpful and are happy to address your further questions.
> > >
> > > **Guarantees on generated samples**
> > >
> > > A central result of [4] provides a bound on the total variation distance between the law of the simulated particles and the target distribution.
> > >
> > > Assume that, for all $n = 1, \ldots, N$,
> > > $$
> > >  {\mathbb{E}}[\|s_{nh} - \nabla \log p_{nh}\|^2]  \le \epsilon_{\mathrm{score}}^2,
> > > $$
> > > where $s$ denotes an approximation of the true score. Let $q_T$ be the density of the particles simulated with this learned score at time $T$. If the step size $h := T/N$ satisfies $h \lesssim 1/L$ with $L \ge 1$, then
> > > $$
> > > \mathrm{TV}(q_T, p_{\mathrm{target}})
> > > \lesssim
> > > \sqrt{\mathrm{KL}(q \,\|\, \gamma^d)\, e^{-T}}
> > > +
> > > (L\sqrt{d}\,h + L m_2 h)\sqrt{T}
> > > +
> > > \epsilon_{\mathrm{score}}\sqrt{T},
> > > $$
> > > (forward process error + discretization error + score error). In our setting, the true score ($-\nabla V(\cdot,t) = \nabla \log p(\cdot,t)$ in our paper) is approximated by
> > >
> > > $$
> > > -\nabla \widetilde{V}(\cdot,t)= \nabla \operatorname{FTT}_n(\cdot),
> > > \qquad t \in [(n-1)h,nh],
> > > $$
> > > where $\operatorname{FTT}_n$ is the functional Tensor Train approximation at time step $n$. Importantly, its gradient can be computed analytically. If
> > >
> > > $$
> > >  {\mathbb{E}}[\| \nabla FTT_n - \nabla \log p_{nh} \|^2] \le \epsilon_{\mathrm{score}}^2 \quad \text{for all } n=1,\ldots,N,
> > > $$
> > >
> > > then the above result from [4] applies directly to our setting.
> > >
> > > Whether this score accuracy is reached depends on the convergence of the learning algorithm. As noted in Remark 2.3, errors may propagate over time, but this effect is small for sufficiently large $N$ and with our stabilization techniques (Remark 3.3). We also outline an additional mitigation strategy in our response to Reviewer LEB8.
> > >
> > > Empirically, our experiments show that $\epsilon_{\mathrm{score}}$ is small enough to yield high-quality samples.
> > >
> > > Further, note that score matching with data is equivalent to a forward KL on path space (Appendix A.9 in [6]), whereas our BSDE-based loss corresponds to a second-moment divergence (Proposition 3.1 in [7]). The optimization principle is similar, but the loss differs because we do not assume access to target samples.
> > >
> > >
> > > **Low-rankness and smoothness**
> > >
> > > We thank the reviewer for the valuable suggestion and include examples illustrating this relationship, drawing on regularity theory for random PDEs in Bayesian inverse problems and explicit constructions. We distinguish two cases of FTT representations of a function $f$:
> > >
> > > (1) $f$ admits an $\epsilon$-approximation  by a FTT with finite rank $\mathbf{r}(ε)$,
> > >
> > > (2) $f$ admits an exact FTT representation with finite rank $\mathbf{r}$.
> > >
> > > (1) *Approximative case:*
> > > Let $f \in L^2(D)$ with $D=[0,1]^d$. Expanding polynomial basis gives
> > > $$f(\mathbf{x}) = \sum_{\nu \in \mathbb{N}^d_0} c_{\nu} \prod_{i=1}^d p_{\nu_i}(x_i). \qquad (F)$$
> > > Smoothness induces decay of $c_{\nu}$, which bounds the FTT rank $(r_1,\ldots,r_{d-1})$ via the TT rank of a truncated coefficient tensor $c_{\nu,\epsilon}$:
> > >
> > > **(A) Anisotropic Mixed Sobolev Smoothness:**
> > > Let $f \in \bigotimes_{i=1}^d H^{s_i}(0,1)$ with $0 < s_1 \le \cdots \le s_d$. Then $|c_{\nu}| \lesssim \prod_{i=1}^d (1+\nu_i)^{-s_i}$. Setting $\beta_1=s_1+s_2$ and $\beta_i=s_i$ for $i\ge2$, then there exists a FTT approximation s.t.
> > > $$
> > > r_i=\lceil \epsilon^{-1/\beta_i} \rceil,
> > > \qquad
> > > \|f-FTT\|_{L^2(D)} \lesssim \sqrt{d}\epsilon.
> > > $$
> > > As $s_i$ increases, $r_i$ stabilizes (e.g., $r_i \to 1$ if $\beta_i \to \infty$).
> > >
> > > **(B) Holomorphic Extension:**
> > > Assume $f$ is holomorphic with polyradii $1<\rho_1\le\cdots\le\rho_d$, then $|c_{\nu}|\lesssim \prod_{i=1}^d \rho_i^{-\nu_i}$. Sparse grids suggest truncation at $\overline{r}_i \sim \ln(1/\epsilon)/\ln(\rho_i)$, i.e. $\nu \in \Lambda=$ {$\nu:\nu_i\le\overline{r}_i$} in $(F)$.
> > >
> > > The matrix rank of the $k$-th unfolding of the tensor $[c_{\nu}]_{\nu\in \Lambda}$ yields
> > >
> > > $$r_k \le \prod_{i=1}^k(\overline{r}_i+1)\lesssim\exp\left(\ln(1/\epsilon)\sum_i\frac{1}{\ln(\rho_i)}\right).$$
> > >
> > > For $\rho_i=i$, $\sum_{i=1}^k 1/\ln\rho_i\sim \ln k$, hence $r_k \lesssim \epsilon^{-\ln k}$.
> > >
> > > (2) *Exact case:*
> > > The function $f(\mathbf{x})=\sin(x_1+\cdots+x_d)$ has FTT rank $(2,\ldots,2)$:
> > > $$
> > > f(\mathbf{x})=
> > > \begin{bmatrix}\sin x_1 & \cos x_1\end{bmatrix}
> > > \prod_{i=2}^{d-1}
> > > \begin{bmatrix}\cos x_i&-\sin x_i\\\sin x_i & \cos x_i\end{bmatrix}
> > > \begin{bmatrix}\cos x_d\\ \sin x_d\end{bmatrix}.
> > > $$
> > > Finally, smoothness is sufficient but not necessary: products $f(\mathbf{x})=\prod_{i=1}^d f_i(x_i)$ with $f_i\in L^2([0,1])$ have minimal rank $(1,\ldots,1)$.
> > >
> > > ---
> > >
> > > Feel free to ask if any further clarification would be helpful. We will add the given discussion and references related on the background of the low-rank discussion (not present here due to space limitations) in the revised version of the paper.
> > >
> > > [6] Berner et al. (2024). An optimal control perspective on diffusion-based generative modeling.
> > >
> > > [7] Sun et al. (2024). Dynamical measure transport and neural PDE solvers for sampling.

---

### Official Review · Reviewer_ZXq6 · 2026-03-13

**Soundness:** 2
**Presentation:** 2
**Significance:** 1
**Originality:** 2
**Overall Recommendation:** 4
**Confidence:** 2

**Summary:**

This paper proposes a new approach for diffusion-based sampling by directly addressing the HJB-type PDE that governs the log-density (and thus the score) needed for reverse-time SDE sampling. To mitigate the high training cost and hyperparameter sensitivity of prior PDE solvers (e.g., PINNs and trajectory-based methods), the authors introduce an FTT (functional tensor train) representation to exploit low-rank structure for efficient high-dimensional function approximation and compression. Combined with a backward-in-time BSDE-inspired iteration, the method aims to deliver faster, more stable, and accurate sampling on challenging target distributions.

**Compliance With Llm Reviewing Policy:**

Affirmed.

**Final Justification:**

I feel my concerens is resolved.

**Key Questions For Authors:**

see weaknessed

**Limitations:**

see weaknessed

**Strengths And Weaknesses:**

Strengths

* Although I am not an expert in PDEs, framing diffusion sampling via an HJB equation and leveraging low-rank approximations to learn/solve it strikes me as a novel and interesting direction.

Weaknesses

* It is unclear to me what concrete advantages this approach provides over the widely used denoising score matching paradigm.
* While the method may indeed be a stronger solver for HJB equations, its motivation and impact in the diffusion-model setting are not fully convincing. In particular, current first-order diffusion (and related flow) models already achieve very strong performance in visual generation.
* The experimental section would benefit from more thorough comparisons to relevant baselines, especially alternative HJB solvers.

---

> ### Author Rebuttal · Authors · 2026-03-26
>
> Dear reviewer ZXq6. Thank you very much for your review. We are happy that you value our novel approach for diffusion-based sampling.
>
> **The claimed advantages over denoising score matching (DSM) are not clearly articulated.**
>
> We believe this concern stems from a mismatch between the problem settings. Denoising score matching fundamentally relies on access to samples from the target. In that setting, one defines a forward "noising" process that gradually transforms data from the target distribution into a tractable prior (typically Gaussian), enabling the learning of the score function along this path. In contrast, our work addresses the classical sampling problem, where no samples from the target are available. Instead, we are given only an unnormalized density $\rho_\mathrm{target}$. This setting is strictly more challenging, as it provides no direct information about the location of high-probability regions (modes) of the distribution (see Figure 1 for an illustration of how our method handles this difficulty).
>
> This distinction is also crucial from a PDE perspective. Our approach leverages the Hamilton-Jacobi-Bellman (HJB) equation, for which the terminal condition is given by the (unnormalized) target density. Since this quantity is explicitly available in our setting, the PDE is well-posed and can be solved numerically. In contrast, in the standard data-driven DSM setting, the target density is not available in closed form, and thus the corresponding boundary condition for the HJB equation is not accessible, rendering this approach inapplicable.
>
> We will highlight this aspect more clearly in the revised version and thank you for raising this aspect.
>
> **The impact in the diffusion model setting (e.g., image generation) is not fully convincing.**
>
> We believe this concern arises from a similar misunderstanding as in the previous point. Our work does not address the standard image generation setting, where samples from the target distribution are available and methods such as diffusion models or flow matching are applicable. Instead, we focus on the substantially more challenging setting in which only an unnormalized target density is given, and no data samples are accessible. In this regime, classical diffusion-based approaches are not directly applicable, as they rely on data to construct the forward noising process or to train score networks. Consequently, the relevant point of comparison is not image-generation pipelines, but rather methods for sampling from unnormalized densities.
>
> Within this setting, we benchmark against a range of alternative approaches, including both learned diffusion-based samplers (Figures 4 and 15) and classical sampling methods (Figure 14). Our results demonstrate that the proposed method achieves superior performance in both computational efficiency and sampling accuracy. In particular, we emphasize that our tensor-train-based sampler attains strong empirical performance while requiring only modest computational resources, highlighting its practical potential in settings where evaluations of the target density are available but data samples are not. We will address all these aspects more clearly in the revised version of the paper and thank you for pointing our potential misunderstandings.
>
> **More relevant baselines, especially alternative HJB solvers, would strengthen the empirical evaluation.**
>
> We would like to emphasize that our experimental section already includes a broad and diverse set of baselines. In particular, we compare against prominent diffusion-based samplers such as PIS (Zhang & Chen, 2022) and DIS (Richter & Berner, 2024) (see Figures 4 and 15), as well as a range of classical and modern sampling methods, including DDS (Vargas et al., 2023), GBS (Blessing et al., 2024), CMCD (Vargas et al., 2024), SCLD (Chen et al., 2024), CRAFT (Matthews et al., 2022), AFT (Arbel et al., 2021), MFVI (Bishop, 2006), and MCD (Doucet et al., 2022). Across these benchmarks, our method consistently outperforms competing approaches in both sampling accuracy and computational efficiency.
>
> Regarding alternative HJB solvers, we point to prior work [1], which explores Physics-Informed Neural Networks (PINNs) for solving the HJB equation. However, as reported in [1], the associated training times are substantially higher than those of our approach, making such methods less practical in our setting. Also, PINNs require the knowledge of the essential domain of the target density and are known to be sensitive to hyperparameter tuning. Furthermore, [1] demonstrates that BSDE-based formulations of the HJB problem are effectively equivalent to the DIS sampler considered above, which is already included in our comparisons.
>
> We are, of course, open to incorporating additional baselines if the reviewer has specific methods in mind that would provide further insight.
>
> [1] Sun, J., et al. (2024). Dynamical measure transport and neural PDE solvers for sampling. arXiv preprint arXiv:2407.07873.

---

> > ### Author Rebuttal · Reviewer_ZXq6 · 2026-04-04
> >
> > I feel my concerens is resolved.

---

### Decision · Program_Chairs · 2026-04-30

**Decision:**

Accept (regular)

**Comment:**

Reviews are positive, though not enthusiastic. Reviews highlight novelty and theoretical soundness. The authors provided a solid rebuttal, that addressed well the reviewers comments. Reviewers indicated that the author's rebuttal resolved their concerns.